# 5'-UTR SNP of *FGF13* causes translational defect and intellectual disability

Xingyu Pan[1,2†], Jingrong Zhao[1†], Zhiying Zhou[3], Jijun Chen[2], Zhenxing Yang[3], Yuxuan Wu[4], Meizhu Bai[4], Yang Jiao[5], Yun Yang[6], Xuye Hu[2,3], Tianling Cheng[1], Qianyun Lu[6], Bin Wang[2,4], Chang-Lin Li[2,3], Ying-Jin Lu[2,3], Lei Diao[4], Yan-Qing Zhong[1], Jing Pan[2], Jianmin Zhu[3], Hua-Sheng Xiao[3], Zi-Long Qiu[1], Jinsong Li[4], Zefeng Wang[6], Jingyi Hui[7], Lan Bao[4,5*], Xu Zhang[1,2,3,5,8*]

[1]Institute of Neuroscience and State Key Laboratory of Neuroscience, CAS Center for Excellence in Brain Science and Intelligence Technology, University of Chinese Academy of Sciences, Chinese Academy of Sciences, Shanghai, China; [2]Shanghai Brain-Intelligence Project Center, Shanghai, China; [3]Shanghai Clinical Center, Chinese Academy of Sciences/Xu-Hui Central Hospital, Shanghai, China; [4]State Key Laboratory of Cell Biology, CAS Center for Excellence in Molecular Cell Science, Shanghai Institute of Biochemistry and Cell Biology, Chinese Academy of Sciences, Shanghai, China; [5]School of Life Science and Technology, Shanghai Tech University, Shanghai, China; [6]CAS-MPG Partner Institute for Computational Biology, Chinese Academy of Sciences, Shanghai, China; [7]State Key Laboratory of Molecular Biology, CAS Center for Excellence in Molecular Cell Science, Shanghai Institute of Biochemistry and Cell Biology, Chinese Academy of Sciences, Shanghai, China; [8]Shanghai Advanced Research Institute, Chinese Academy of Sciences, Shanghai, China

**\*For correspondence:**
baolan@sibcb.ac.cn (LB);
xu.zhang@ion.ac.cn (XZ)

[†]These authors contributed equally to this work

**Competing interests:** The authors declare that no competing interests exist.

**Abstract** The congenital intellectual disability (ID)-causing gene mutations remain largely unclear, although many genetic variations might relate to ID. We screened gene mutations in Chinese Han children suffering from severe ID and found a single-nucleotide polymorphism (SNP) in the 5'-untranslated region (5'-UTR) of fibroblast growth factor 13 (FGF13) mRNA (NM_001139500.1:c.-32c>G) shared by three male children. In both HEK293 cells and patient-derived induced pluripotent stem cells, this SNP reduced the translation of FGF13, which stabilizes microtubules in developing neurons. Mice carrying the homologous point mutation in 5'-UTR of *Fgf13* showed delayed neuronal migration during cortical development, and weakened learning and memory. Furthermore, this SNP reduced the interaction between *FGF13* 5'-UTR and polypyrimidine-tract-binding protein 2 (PTBP2), which was required for *FGF13* translation in cortical neurons. Thus, this 5'-UTR SNP of *FGF13* interferes with the translational process of *FGF13* and causes deficits in brain development and cognitive functions.

## Introduction

Intellectual disability (ID) is defined as an overall intelligence quotient (IQ) of less than 70, together with a disability characterized by 'significant limitations both in intellectual functioning and in adaptive behavior as expressed in conceptual, social and practical skills' with onset before the age of 18 years (*Dsm, 1994*). It is estimated that 1–3% of the general population suffers from ID, which is caused by either genetic or environmental factors during development (*Larson et al., 2001*; *Ropers, 2008*). About 5–10% of male ID patients are caused by genetic variations in X chromosome, including structure variations (SVs), copy number variations (CNVs), and single-nucleotide variations

(SNVs) (*Lubs et al., 2012*), and named as X-linked intellectual disability (XLID) (*Lehrke, 1972*). Large-scale and systematic sequencing for ID individuals and related families has markedly accelerated the study of the intact genetic spectrum for XLID, and about 200 candidate genes have been suggested for XLID (*Gécz et al., 2009*; *Stevenson and Schwartz, 2009*). The sequencing of coding exons of 718 X-chromosome genes from 208 families with ID patients further identifies 9 novel XLID-related genes (*Tarpey et al., 2009*), and yet only covers 65% regions of these X-chromosome genes but does not include noncoding sequences. Whether the ID-related genetic variations cause ID pathogenesis needs to be proved with cell models and animal models. In addition, huge difference of genetic background exists between East Asian (EAS) and European (*Sirugo et al., 2019*), while populations in the Europe and the USA have magnificent databases of human genomics. Therefore, the investigation of genetic mutations in the genetic spectrum of Chinese population is of significance.

The proper translation of mRNAs is critical for cellular functions, and neurons particularly rely on the accurately controlled mRNA translation both spatially and temporally. Defects in the translational control have been linked to numerous neurodegenerative disorders (*Darnell and Klann, 2013*; *Gkogkas et al., 2013*; *Buffington et al., 2014*). The 5′-untranslated region (5′-UTR) of mRNA is the frame for 43S subunit to scan along at the initiation of translation, and features within the 5′-UTR could modulate the efficiency of mRNA translation, including 5′-UTR length, secondary structures, multiple upstream open reading frames (uORFs), and internal ribosome entry site (IRES) (*Chatterjee and Pal, 2009*). Moreover, various elements located in the UTRs of mRNAs influence the translation and may cause diseases under altered conditions (*Kondo et al., 1998*; *Tassone et al., 2001*; *Scheper et al., 2007*; *Wang et al., 2007*). For example, the fragile X syndrome (FXS) is the most frequent form of inherited XLID caused by an expansion of the CGG triplet repeat on X chromosome within the 5′-UTR of *FMR1*, which leads to the excessive methylation and silenced *FMR1* transcription (*Bell et al., 1991*; *Oberlé et al., 1991*; *Verkerk et al., 1991*; *Yu et al., 1991*). The current studies are biased to identify coding variants implicated in ID (*Gilissen et al., 2014*), while the pathogenic non-coding regulatory variants, including duplication, insertion, and SNVs, have only been discovered in a few cases to date (*Borck et al., 2006*; *Bonnet et al., 2012*; *Huang et al., 2012*; *Kumar et al., 2016*). Nevertheless, the deleterious functions and underlying mechanisms of those noncoding variants remain to be further explored.

The regulation of cytoskeleton dynamics is important for neuronal morphogenesis, and several microtubule-stabilizing proteins (MSPs), such as doublecortin identified from human lissencephaly patients (*Gleeson et al., 1998*), are involved in the process of axon branching (*Kappeler et al., 2006*) and neuronal migration (*Gleeson et al., 1999*; *Koizumi et al., 2006*). Fibroblast growth factor 13 (FGF13), a nonsecretory fibroblast growth factor (FGF), belongs to the *FGF11* subfamily (*Goldfarb, 2005*; *Zhang et al., 2012*). *FGF13* might be a candidate gene for the syndromal and nonspecific forms of XLID mapped to the q26 region of X-chromosome (*Smallwood et al., 1996*; *Gecz et al., 1999*; *Goldfarb, 2005*). In a Börjeson–Forssman–Lehmann syndrome-like patient, *FGF13* was interrupted by a duplication breakpoint (*Gecz et al., 1999*). Moreover, FGF13b, the cytoplasmic isoform of FGF13, serves as a MSP required for axon development, neuronal migration, and brain development, and FGF13-deficient mice exhibit impaired learning and memory (*Wu et al., 2012*). Genetic disruption of *FGF13* due to chromosome translocation in a genetic epilepsy family is also associated with cognitive impairment, in which not only the proband but also his mother's maternal aunt together with her two children all exhibit cognitive impairments (*Puranam et al., 2015*). However, the function and the regulating mechanisms of *FGF13* mutations in ID still lack systematic investigations.

Here, we identified a single-nucleotide polymorphism (SNP) in the 5′-UTR of *FGF13* of 3 severe ID children (IQ < 40). This 5′-UTR mutation selectively hampered the protein translation of FGF13. The defects of neuronal migration and axon branching along with ID-related behaviors appeared in mice carrying the homologous point mutation of FGF13. This point mutation impaired the binding of *FGF13* 5′-UTR with polypyrimidine tract-binding protein (PTBP)1/2, which resulted in the translation dysfunction of *FGF13*. Thus, our study provides a novel SNP participating in the dysfunction of protein synthesis and pathogenesis of ID patients, and reveals both molecular and developmental mechanisms by which the 5′-UTR mutation in *FGF13* impairs cognitive functions.

## Results

### Sequencing of Chinese ID children reveals a novel SNP in *FGF13*

To collect the specific ID risk mutations of Chinese Han population, we performed Sanger sequencing of *FGF13* on 100 severe ID children (IQ < 40) because of an important role of FGF13 in regulating brain development. The full-scale IQ scores were detected by the test of Wechsler Intelligence Scale for Children. Among them, three male cases (allele frequency = 2.2%) shared an identical cytosine-to-guanine substitution (C>G) in the 5′-UTR of *FGF13* (NC_000023.10:g.138286301g>C, NM_001139500.1:c.-32c>G) (*Figure 1A*). The SNP identified in the present study was not reported in previous SNP database at that time (dbSNP, 1000G), but the new build of 1000G database in 2015 (rs757803941) included this site, where the SNP was found in one heterozygous individual from Chinese Han South population (CHS). Meanwhile, the EAS population of ExAC database also reported this SNP in 2016 and showed that it is an EAS-specific SNP (*Exome Aggregation Consortium et al., 2016*). In detail, the mutation is located at the position of chrX: 138,286,301 (hg19) of *FGF13* with serial number of rs757803941 in dbSNP. Up to now, the allele frequency of this SNP is 1/764 (0.13%) in EAS population of 1000G, 15/6599 (0.23%) in EAS population of ExAC, 26/32416 (0.080%) in Asian population of gnomAD, and 9/2910 (0.31%) in Korean population from KRGDB database. A recent paper reported a large-scale whole-genome sequencing data of Chinese population (the China Metabolic Analytics Project [ChinaMAP]) (*ChinaMAP Consortium et al., 2020*), and this SNP exhibits allele frequency of 0.23%. Therefore, these data suggest that a novel specific SNP in *FGF13* among EAS population is found in severe ID children.

As part of further efforts, we performed the target region capture-sequencing that covered 262 known ID-related genes (see *Supplementary file 1*) in three boys and their parents to screen the pathogenic SNPs and small insertions and deletions (In-Dels) occurred in ID-related genes (see Materials and methods). As expected, in each of these three families, the SNP of *FGF13* is inherited from the mother, which is in accordance with the presumed mode of X-linked recessive inheritance (*Figure 1B*). After annotated by the clinVar database (see Materials and methods), the SNPs located in the ID-related genes were predicted to be pathogenic or not (see *Supplementary file 2*). Notably, the father of family 1 had one SNP suggested as pathogenic, and the mother of this family also had one pathogenic SNP. However, their son did not inherit these two pathogenic SNPs from parents and did not have other pathogenic ID-related SNPs. None of the pathogenic ID-related SNPs were found in other two families. The members of all three families did not carry any pathogenic ID-related In-Dels (see *Supplementary file 2*).

Furthermore, to search for the genomic characteristics of these three children, we performed whole-genome sequencing. After analysis of all SNPs, In-Dels, and CNVs in their whole genome (see *Supplementary file 3*), we found 3655/3687/3564 SNPs located in 1156 ID-related genes suggested by Gilissen and colleagues (*Gilissen et al., 2014*). When compared with the pathogenicity of the SNPs through annotation with the clinVar database (see *Supplementary file 3*), only one heterozygous SNP in *RAF1* from child 2 was suggested to be pathogenic and related to cardiomyopathy dilated. We found that none of the In-Dels and CNVs located in the ID-related genes was crossmatched to be pathogenic. Thus, these results suggest that the *FGF13* SNP found in these ID children may be a dominant pathogenic genetic variation.

Three boys carrying this SNP were also examined in clinic. Children 1 and 3 displayed mild to profound defects in adaptive behaviors through the test of Vineland Adaptive Behavior Scales. They all showed problems in speaking, and children 1 and 2 could only learn to make simple monosyllabic sounds, while child 3 was milder and could go to normal primary school at first and second grades after training. In addition, children 1 and 2 lacked self-care ability for defecation and child 3 was all self-care. Moreover, children 1 and 3 had high pain threshold and were suspected of mild autism. Abnormalities of motor development were not obviously observed in all three children. Other clinical features and developmental histories of these three ID children were also recorded (see *Supplementary file 4*).

### Point mutation in human 5′-UTR impairs *FGF13* translation

We first analyzed the location of *FGF13* 5′-UTR variation and its correlated transcript variants. It locates at the exon 2 of *FGF13*, which is involved in the transcription of transcript variants 2–4, and

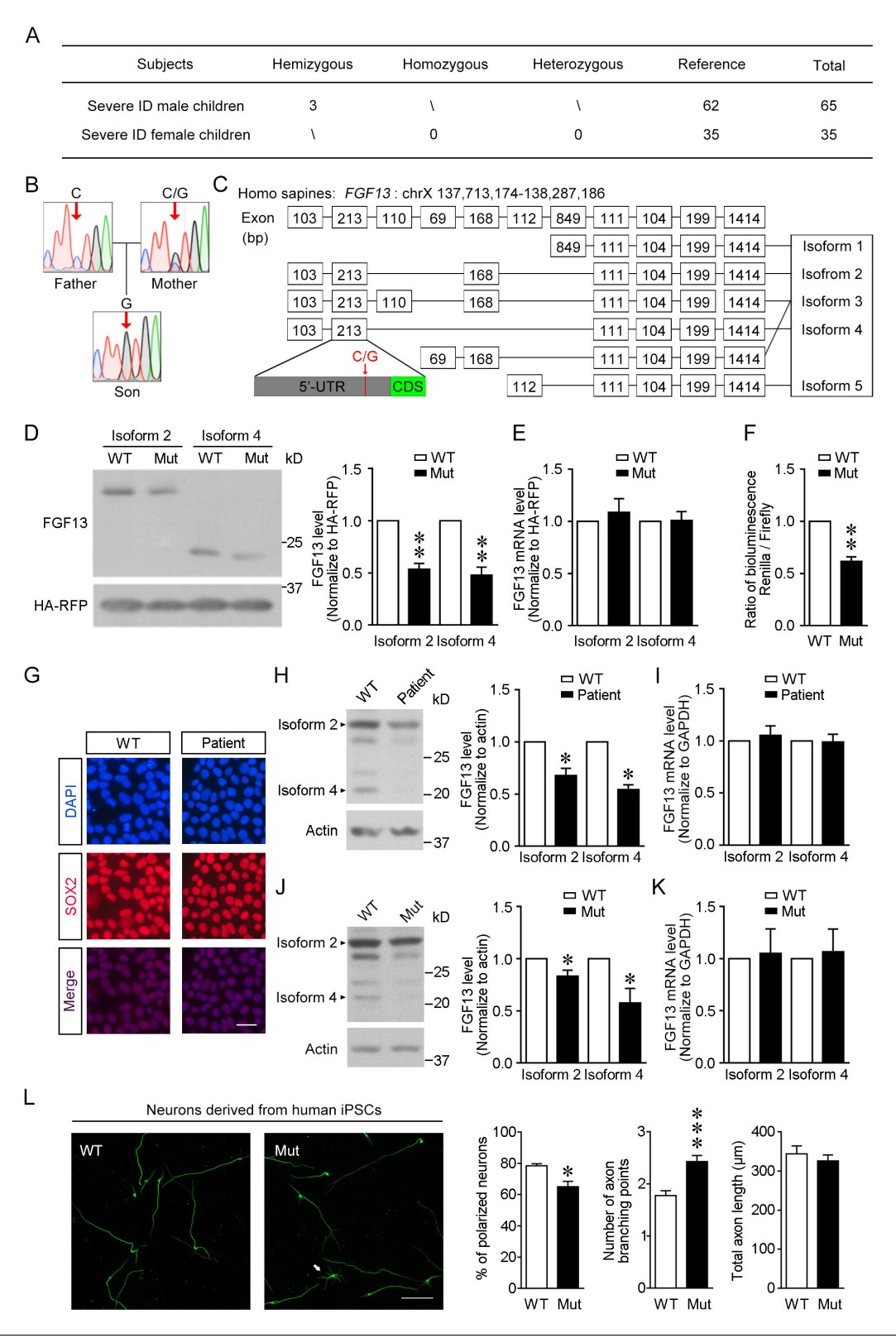

**Figure 1.** The newly identified single-nucleotide polymorphism (SNP) in *FGF13* impairs the protein translation. (**A**) The number of subjects involved in the present study is listed. A sum of 100 severe intellectual disability (ID) children (intelligence quotient [IQ] < 40) with detailed clinical histories were tested for the SNP of *FGF13*. (**B**) Sanger sequencing results of 1 ID boy with his parents revealed X-linked recessive inheritance model of the 5′-untranslated region (5′-UTR) SNP of *FGF13*. One of all three families is shown here, and in each family the son carried the SNP, while the mother

*Figure 1 continued on next page*

*Figure 1 continued*

showed heterozygous base in this site and the father showed normal base. (C) The point mutation (indicated by red arrow) is located at the 5′-UTR of *FGF13* exon 2 and is involved in the expression of three FGF13 isoforms. (D, E) The 5′-UTR SNP of *FGF13* reduced the protein levels of isoforms 2 and 4 of FGF13 in HEK293 cells. Immunoblotting results of FGF13 and HA-RFP are shown, and HA-RFP was used as a normalized control for the transfection of plasmids. The protein (D) and mRNA (E) levels of FGF13 isoforms 2 and 4 were examined, and the mutant *FGF13* 5′-UTR (Mut) reduced the protein levels of FGF13 isoform 2 (n = 4, p=0.0037) and isoform 4 (n = 4, p=0.0067), but did not alter their mRNA levels, compared to wildtype (WT) (isoform 2: n = 6, p=0.5364; isoform 4: n = 6, p=0.9306). (F) The activity of 5′-UTR was assessed in the dual-luciferase reporter assay, which was indicated by the luminescence ratio of Renilla and Firefly. The Mut 5′-UTR attenuated the reporter Renilla translation (n = 4, p=0.0032). (G) Representative images of DAPI and SOX2 immunostaining show the induced pluripotent stem cells (iPSCs) derived from the renal epithelial cells of ID patients and controls. Scale bar, 100 μm. (H) Immunoblotting showed that the expression of FGF13 isoforms was downregulated in the ID patient-derived iPSCs compared to WT iPSCs (isoform 2: n = 3, p=0.0437; isoform 4: n = 3, p=0.0101). Arrowheads indicate bands of FGF13 isoforms 2 and 4. (I) Quantitative real-time PCR shows that the FGF13 mRNA level was not decreased in the ID patient-derived iPSCs (isoform 2: n = 3, p=0.6136; isoform 4: n = 3, p=0.8951). (J) Immunoblotting shows that the introduction of the mutant site (C>G) in *FGF13* 5′-UTR in WT iPSCs could reduce the expression of FGF13 isoforms 2 and 4 (isoform 2: n = 3, p=0.0497; isoform 4: n = 3, p=0.0464). Arrowheads indicate bands of FGF13 isoforms 2 and 4. (K) Quantitative real-time PCR shows that the FGF13 mRNA level was not altered when WT iPSCs were transformed to Mut ones (isoform 2: n = 3, p=0.8527; isoform 4: n = 3, p=0.7903). (L) Immunostaining of Tuj1 (green) shows that the induced neurons derived from iPSCs carrying the 5′-UTR SNP of *FGF13* displayed a lower ratio of polarized neurons (n = 3 experiments, p=0.0500) and increased axon branching points with unchanged total axon length (n = 120 and 131 for WT and Mut neurons, respectively, from three experiments, p=5.45E-05 for axon branching points and p=0.4409 for total axon length). Unpolarized neuron is labeled by white arrow in the Mut group. Scale bar, 100 μm. *p<0.05 and ***p<0.001 versus WT by Mann–Whitney U test. Data are represented as mean ± SEM (D–F, H–L). *p<0.05 and **p<0.01 versus WT by Student's *t*-test (D, F, H, J).

The online version of this article includes the following source data and figure supplement(s) for figure 1:

**Source data 1.** Source data for *Figure 1*.

**Figure supplement 1.** Manipulation of the *FGF13* 5′-untranslated region (5′-UTR) in induced pluripotent stem cells (iPSCs) with the CRISPR/Cas9 editing.

**Figure supplement 1—source data 1.** Source data for *Figure 1—figure supplement 1*.

the translation of isoforms 2–4 of FGF13 (*Figure 1C*). To verify the expression pattern of FGF13 in human brain development, we first reanalyzed the published single-cell database of human fetal pre-frontal cortex (*Zhong et al., 2018*) and found that FGF13 mRNA was abundant in both neural progenitor cells (NPCs) and neurons (*Figure 1—figure supplement 1A*). Then, the real-time PCR showed that in human fetal brain the transcript variants 2 and 4 (NM_001139500.1 and NM_001139498.1) that might be influenced by the *FGF13* SNP were highly expressed and more enriched compared to others (*Figure 1—figure supplement 1B*).

To test the role of the *FGF13* 5′-UTR variation, we constructed pcDNA3.1 plasmids expressing the transcript variant 2 or 4 of *FGF13*, along with the whole 5′-UTR of human sequence origins to be inserted before the coding sequence, which were able to be translated into the isoform 2 or 4 of FGF13, respectively (*Figure 1D*). A HA tag-fused RFP (HA-RFP) was also co-expressed in the pcDNA3.1 plasmid to normalize the expression level due to different transfection efficacy. Importantly, immunoblotting showed that the protein levels, but not the mRNA levels, of both isoforms 2 and 4 of FGF13 were reduced in HEK293 cells expressing the plasmids carrying the single-nucleotide mutation of *FGF13* compared to that of wildtype (*Figure 1D, E*). Furthermore, a dual-luciferase reporter assay also confirmed that the mutant 5′-UTR of *FGF13* significantly reduced the reporter protein expression compared to the wildtype 5′-UTR (*Figure 1F*). Taken together, these results suggest that the point mutation in *FGF13* 5′-UTR impairs the translation rather than the transcription procedure.

## Endogenous FGF13 expression and neuronal morphology are impaired in human cells carrying the 5′-UTR SNP of *FGF13*

The induced pluripotent stem cells (iPSCs) derived from patients usually exhibit cellular defects, which provides insight into the disease pathophysiology (*Grskovic et al., 2011*). To uncover the phenotypes of human cells carrying the 5′-UTR point mutation of *FGF13*, we generated iPSCs from urothelial cells of three ID children and three normal children through episomal reprogramming (*Grskovic et al., 2011*). All iPSCs were positive for the progenitor markers SOX2 (*Figure 1G*) and *NANOG* (*Figure 1—figure supplement 1C*), but negative for the differential cell markers *PAX6*, *MAP2*, *GFAP*, and *NES* (*Figure 1—figure supplement 1C*). Notably, the protein levels of both isoform 2 and isoform 4 of FGF13 were significantly reduced in patient-derived iPSCs (*Figure 1H*),

whereas the mRNA levels remained unchanged (*Figure 1I*). Besides the bands of FGF13 isoforms 2 and 4, other FGF13 bands, which might represent FGF13 isoform 3 or FGF13 isoforms with post-translational modifications, were also verified by the antibody absorption experiment. The intensity of these FGF13-immunoreactive bands exhibited similar reduction in the patient-derived iPSCs. Thus, the protein level of endogenous FGF13 is decreased in patient-derived iPSCs.

In order to confirm the specific effect of this 5'-UTR point mutation in FGF13 and exclude the variation in FGF13 expression due to individual genetic background, we used the CRISPR/Cas9 system to introduce the mutant nucleotide (C>G) into human iPSCs (*Figure 1—figure supplement 1D*). We obtained precisely editing three control-mutated iPSCs (*Figure 1—figure supplement 1E*) after sequencing the top 10 ranked potential off-target sites of the designed sgRNA (*Figure 1—figure supplement 1F*). Compared to the unediting iPSCs, a single-nucleotide alteration (C>G) for human iPSCs efficiently reduced the protein levels of both FGF13 isoform 2 and isoform 4 even if the mRNA levels of transcript variants 2 and 4 were not altered (*Figure 1J, K*). These data suggest that the 5'-UTR point mutation of FGF13 specifically attenuates *FGF13* translation in human iPSCs.

We further test whether the 5'-UTR point mutation of FGF13 results in the morphological alteration of human neurons. Neurons were induced through Cas9 editing from control or mutant NPCs that were differentiated from iPSCs (*Zhang et al., 2016*), and then we examined the polarization and axon development of these neurons. Induced neurons were positive for the neuron-specific marker Tuj1 (*Figure 1L*, *Figure 1—figure supplement 1G*). Immunostaining of Tuj1 showed a decreased ratio of polarized neurons and an increased axon branches with unchanged total axon length in the mutant neurons compared to control neurons (*Figure 1L*, *Figure 1—figure supplement 1G*). Taken together, these results verify the pathological role of the 5'-UTR point mutation of *FGF13* in a human cellular model, in which the mutation reduces endogenous protein translation and impairs neuronal axon formation.

## Homologous point mutation in the 5'-UTR of *Fgf13* in the mouse impairs *Fgf13* translation

Animal models are usually used to study the pathological role of genetic variations at organism level; therefore, we further tested the role of this mutation in mutant mice. The human *FGF13* sequence was highly conserved among many species including the mouse (*Figure 2A*), then we constructed the homologous mouse 5'-UTR sequence of *Fgf13* into the pcDNA3.1 plasmid to test whether this mutation could similarly reduce the protein level of FGF13. Full-length sequence of *Fgf13* 5'-UTR was inserted before the coding sequences of mouse FGF13 isoform 2 or 4 in the pcDNA3.1 plasmid that co-expressed a HA-RFP. Consistently, immunoblotting showed that the 5'-UTR point mutation of *Fgf13* also attenuated the protein expression of FGF13 isoforms 2 and 4 (*Figure 2B*), but did not alter their mRNA levels in HEK293 cells (*Figure 2—figure supplement 1A*). Similarly, the dual-luciferase assay also detected that the 5'-UTR point mutation of *Fgf13* reduced the reporter protein expression (*Figure 2C*). Thus, the homologous 5'-UTR point mutation of *Fgf13* in the mouse also impairs *Fgf13* translation.

To create the knock-in mice with the homologous 5'-UTR point mutation (C>G) in *Fgf13*, we used the CRISPR/Cas9 system to introduce the single-nucleotide mutation (*Figure 2—figure supplement 1B, C*). After examining 10 top-ranked sequences with potential off-target of the designed sgRNA, we excluded potential 'off-target' effects at the predicted sites in the mutant mice (*Figure 2—figure supplement 1D*). The expression of FGF13 isoforms 2 and 4 was examined at postnatal day 0 (P0) of both mutant and littermate control mice. Notably, the transcripts 2 and 4 were enriched in the P0 mouse brain, and the protein levels of FGF13 isoforms 2 and 4 in both cerebral cortex and hippocampus of mutant mice were lower than that in control mice, while the mRNA levels were not changed (*Figure 2D*, *Figure 2—figure supplement 1E, F*). Therefore, the homologous 5'-UTR point mutation induced in the mouse *Fgf13* gene also impairs the protein translation in the mouse brain.

## Learning and memory are impaired in *Fgf13* mutant mice

Behavior tests of the male *Fgf13* mutant mice were performed to help us understand the integral phenotypes in intellectual development, which are the major defects in ID patients. Morris water maze was used to detect the spatial learning and memory of mice. Most control littermate mice learned to use distal visual cues to navigate themselves to the hidden platform during a 6-day

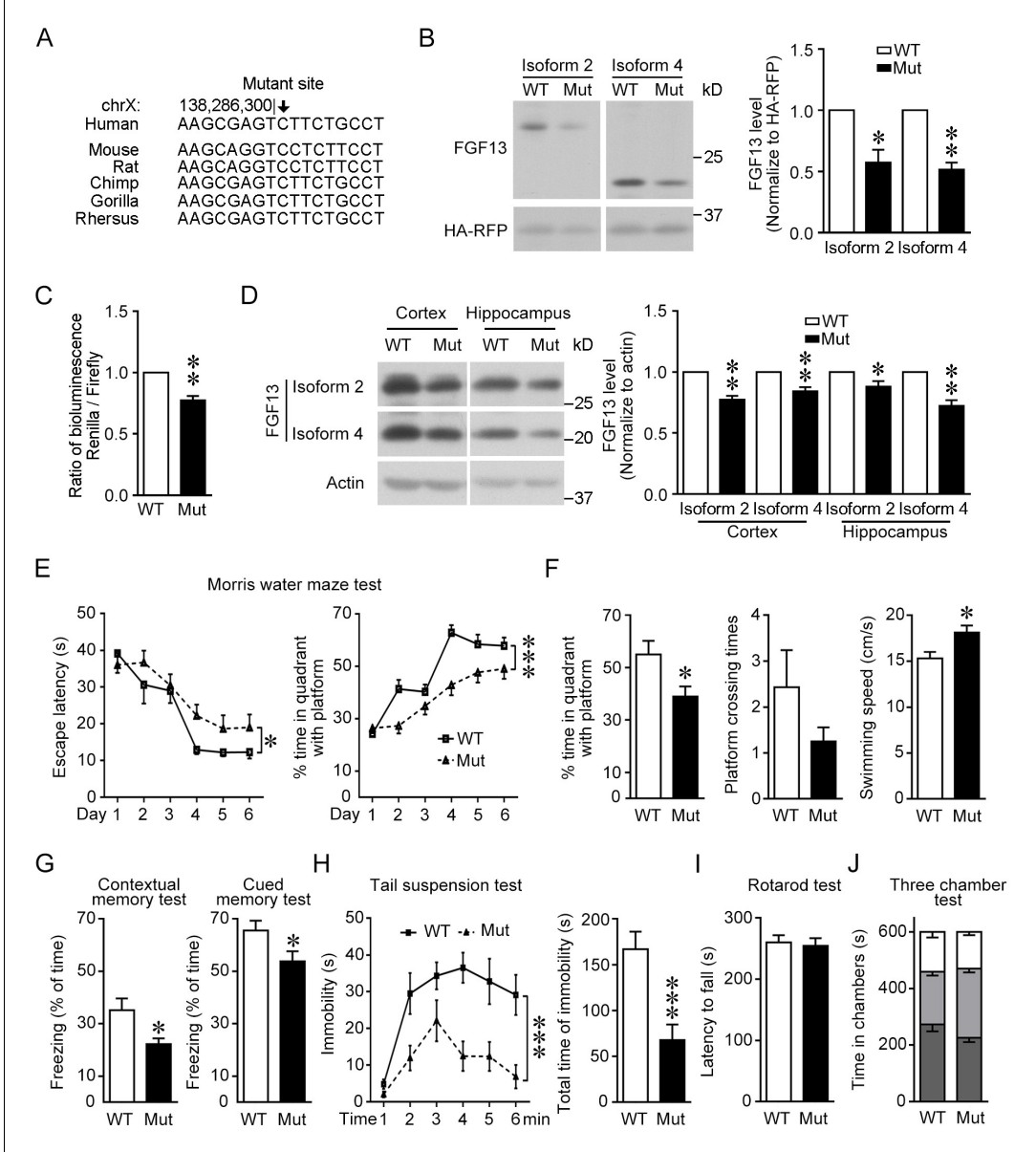

**Figure 2.** Mice with the point mutation in *Fgf13* 5′-untranslated region (5′-UTR) reduce protein translation and exhibit impaired learning and memory. (A) The region around the mutation site is highly conserved among species. The mutation site is highlighted with the arrow and remains constant in various species. (B) The homology point mutation in the 5′-UTR of *Fgf13* in the mutant (Mut) mouse could reduce the translation of FGF13 isoforms in HEK293 cells (isoform 2: n = 4, p=0.0263; isoform 4: n = 4, p=0.0038). The two FGF13 isoforms were immunoblotted in the same membrane but not adjacent. (C) Dual-luciferase reporter assay verified that the homologous mutation in *Fgf13* 5′-UTR impaired the reporter Renilla translation (n = 6, p=0.0013). (D) The expression of FGF13 isoforms was impaired in the cerebral cortex and the hippocampus of *Fgf13* 5′-UTR Mut mice compared to that of the wildtype (WT) mice at P0 (n = 5, p=0.0017 and p=0.0098 for cortex; n = 5, p=0.0286 and p=0.0021 for hippocampus). Bands of FGF13 isoforms 2 and 4 were immunoblotted in the same membrane. (E) The escape latency and performance of the *Fgf13* 5′-UTR Mut mice in Morris water maze were impaired compared to that of the WT mice during 6-day training period. The performance was indicated by the time spent in the quadrant with the platform and the escape latency to reach the platform (WT: n = 7; Mut: n = 12; left, p=0.0332; right, p=1.23E-05). *p<0.05 and ***p<0.001 versus WT by two-way ANOVA. (F) The Mut mice showed impaired spatial memory in the probe test day, which was indicated by the time spent in the quandrant with platform (p=0.0301) and the times crossing the platform position (p=0.2748 by Mann–Whitney U test). Moreover, the average swimming speed (p=0.0161) of Mut mice (n = 12) was higher than that of WT mice (n = 7). (G) Associate memory of Mut mice (n = 14) in short-term contextual or cued memory tests was impaired as compared to that of WT mice (n = 10), indicated by the freezing time in the chambers (p=0.0227 contextual and p=0.0405 for cued test). (H) Performance in the tail suspension test for Mut mice was quantitatively analyzed. The Mut mice (n = 14) showed less depression indicated by the reduced total time of immobility than WT mice (n = 10). Left, p=3.36E-10 (***) versus WT by two-way ANOVA; right, p=0.0009 (***) versus WT by Mann–Whitney U test. (I) Rotarod experiment showed that the motor coordinate function of Mut mice (n = 14) was not

*Figure 2 continued on next page*

*Figure 2 continued*

altered (p=0.8076 by Mann–Whitney U test) compared to that of WT mice (n = 10). (J) The results of three-chamber test showed that the social interaction of Mut mice was not altered compared to that of WT mice (WT: n = 10; Mut, n = 14; p>0.9999 by two-way ANOVA). White box stands for time in chamber of inanimate object, while gray box represents the time in center zone and dark gray box indicates the time sniffing another mouse in the chamber. Data are represented as mean ± SEM (**B–J**). *p<0.05 and **p<0.01 versus WT by Student's *t*-test (**B–D, F, G**).

The online version of this article includes the following source data and figure supplement(s) for figure 2:

**Source data 1.** Source data for *Figure 2*.
**Figure supplement 1.** Construction of mice with the point mutation in *Fgf13* 5'-untranslated region (5'-UTR) using CRISPR/Cas9 system.
**Figure supplement 1—source data 1.** Source data for *Figure 2—figure supplement 1*.

training period, whereas the mutant mice showed increased escape latency and reduced time spent in target quadrant during the training process (*Figure 2E*). In the probe test to assess spatial memory, mutant mice showed a less preference for target quadrant than control mice (*Figure 2F*). Meanwhile, the swimming speed of mutant mice was even increased compared to that of control mice, excluding the possibility that the longer escape latency was caused by swimming disability (*Figure 2F*). Furthermore, in the fear conditioning test, mutant mice displayed impaired associated memory revealed by a reduction of freezing time in both contextual and cued memory tests (*Figure 2G*). Thus, learning and memory are impaired in *Fgf13* mutant mice.

Moreover, mutant mice displayed decreased depression-like behavior, which was revealed by a reduction of immobility time in the tail suspension test (*Figure 2H*). Meanwhile, mutant mice did not show apparent alteration of motor ability in the rotarod test and the social interaction behavior in the three-chamber test (*Figure 2I, J*). The anxiety-related behavior in the light-dark box test (*Figure 2—figure supplement 1G*), the capacity to recognize novel object in the novel object recognition test (*Figure 2—figure supplement 1H*), and the spontaneous activity in the open field test of mutant mice (*Figure 2—figure supplement 1I*) all remained normal. These results suggest that the *Fgf13* mutant mice also display mood disorders at some aspects.

### *Fgf13* mutant mice exhibit delayed neuronal migration, increased axon branching, and reduced spine density

Since the point mutation reduced the protein expression of FGF13 and impaired the morphology in human iPSC-derived mutant neurons, we then tested whether mutant mice also showed defects in neuronal development. The behavioral changes of mutant mice provide a possibility that the cortical development is affected by the point mutation of *Fgf13*. To test whether neuronal migration was changed in mutant mice, we used in utero electroporation to transfect plasmids at E14.5 in the radial glial progenitors that could be differentiated into radial migrating neurons in later developmental stages. Immunostaining showed that the percentage of neurons remained in the white matter or cortical layers V–VI of mutant mice was higher than those in control mice at P0, and the migration delay was kept till P7 (*Figure 3A, B*). When the plasmid expressing wildtype *Fgf13* was co-transfected, the migratory defect of mutant mice could be partially rescued by addition of FGF13 (*Figure 3B*), indicating a critical role of FGF13 in regulating neuronal migration. Notably, the vast majority of cortical neurons in both control mice and mutant mice migrated into layers II–IV at P14 (*Figure 3—figure supplement 1A, B*), suggesting potential compensation in neuronal migration in mutant mice at late phase of development. To further study the laminar defects in the cerebral cortex of mutant mice, brain sections were labeled with Cux1 to mark the superficial cortical neurons (*Figure 3—figure supplement 1C*). In mutant mice, Cux1-positive neurons were distributed in both layers II–IV and mislocated throughout layers V–VI, with reduced thickness of superficial cortical layer at P0 but compensated at P7 (*Figure 3—figure supplement 1D*). These data suggest that the decreased FGF13 protein in the mutant mice is sufficient to cause delayed positioning of neurons during the development of cerebral cortex.

To further clearly confirm the role of *FGF13* 5'-UTR SNP in regulating neuronal morphology, we stained cortical neurons cultured from wildtype littermates and *Fgf13* mutant mice for 4 days with Tuj1 to analyze the neuronal morphology. Most neurons from wildtype mice extended a unique primary axon, while nearly 40% of neurons from mutant mice did not develop into polarized neurons (*Figure 3C, D*). The total number of axon branches was also increased in the neurons from mutant

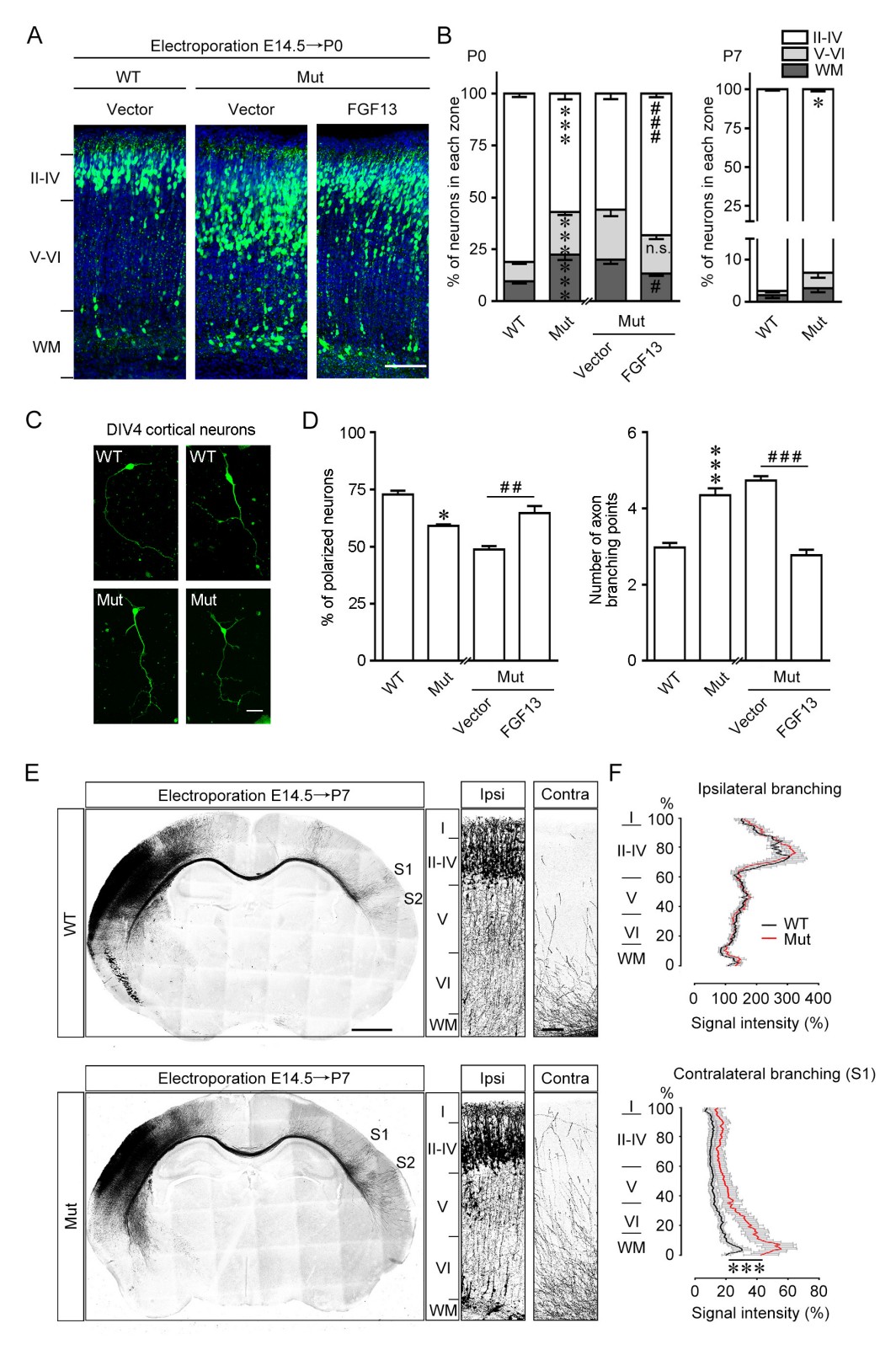

**Figure 3.** *Fgf13* point mutation mice show delayed radial migration and increased axon branching. (**A, B**) Representative images and quantitative analysis of the cerebral cortex at P0 in *Fgf13* point mutant (Mut) mice electroporated with pcDNA3.1 vector at E14.5 showed that the radial migration of cortical neurons was delayed, and such a delay of neuronal migration was sustained till P7, compared to that of wildtype (WT) mice. Re-expression of FGF13 could rescue the migratory defects. Left of (**B**), n = 15, 19, 26, and 31 brain slices from 3 to 5 WT or Mut littermate mice. Layers II–IV: p=3.23E-08

*Figure 3 continued on next page*

*Figure 3 continued*

(***) versus WT, while p=0.0005 (###) versus Mut plus vector; layers V–VI: p=5.47E-07 (***) versus WT, while p=0.3193 (n.s.: not significant) versus Mut plus vector; WM: p=0.0006 (***) versus WT, while p=0.0121 (#) versus Mut plus vector. Right of (**B**), n = 9 and 12 brain slices from 3 to 4 WT or Mut littermate mice. Layers II–IV: p=0.0271 (*) versus WT; layers V–VI: p=0.2388; WM: p=0.2342. Scale bar, 100 μm (**A**). All statistical analyses were performed by Mann–Whitney U test. (**C, D**) DIV4 cortical neurons cultured from Mut mice showed lower ratio of polarized morphology and displayed more axon branches compared to that from WT mice (n = 179 neurons for WT and 160 neurons for Mut from three experiments). Neurons were immunostained with Tau. Such changes were rescued by re-expression of FGF13 (n = 295 neurons for Mut plus vector and 127 neurons for Mut plus FGF13 from five experiments). p=0.0500 (*) and p=8.78E-09 (***) versus WT, while p=0.0040 (##) and p<1.00E-15 (###) versus indicated group by Mann–Whitney U test. Scale bar, 50 μm (**C**). (**E**) Representative images of P7 brain slices in mice electroporated with green fluorescent protein (GFP) plasmid at E14.5 show the distribution of axon projection in the cerebral cortex. Right panels show magnification of bilateral primary somatosensory cortex. The Mut mice showed that the axon arbors were more complex, and the axons could project to both the contralateral S1 and S2 cortical regions, while in WT mice the axons mainly reached the contralateral S1 cortical region. Scale bars, 1000 (left) and 100 (right) μm. (**F**) Quantitative analysis of axon arbors of bilateral somatosensory cortex showed increased axon branching in the contralateral hemisphere in Mut mice. The axon intensity in the ipsilateral layer V was used for normalization. The Mut mice showed more complex and intense axon projections in the contralateral side (n = 5 brain slices from three WT mice, and n = 8 slices from three Mut littermate mice). p<1.00E-15 (***) versus WT in contralateral branching by two-way ANOVA. Data are represented as the mean ± SEM (**B, D, F**).

The online version of this article includes the following source data and figure supplement(s) for figure 3:

**Source data 1.** Source data for *Figure 3*.
**Figure supplement 1.** Alterations of neuronal migration, axon projection, and spine density in the brain of *Fgf13* 5'-untranslated region (5'-UTR) mutant (Mut) mice.
**Figure supplement 1—source data 1.** Source data for *Figure 3—figure supplement 1*.

mice (*Figure 3D*). Furthermore, the abnormal neuronal polarization and axon branching of mutant mice were able to be reversed by the expression of FGF13, indicating a critical role of FGF13 in regulating axon branching (*Figure 3D*). Therefore, the homologous 5'-UTR point mutation of *Fgf13* in the mouse impairs the axon development of neurons.

Moreover, we detected the axon projection of neurons to verify the effect of the point mutation on the complexity of axon arbors in developing mouse brain in vivo. Heterozygous female mice were in utero eletroporated with GFP plasmids at E14.5 to label the radial glial progenitors and callosal axons crossing the corpus callosum (*Figure 3E*). The axons reached contralateral primary somatosensory cortex (S1) and began to branch into different cortical layers at P7, the time for the peak FGF13 expression (*Wu et al., 2012*). Ipsilateral axon branching was not altered in the layer V and thus used to normalize transfection variation (*Figure 3F*). However, axon arbors projected to the contralateral somatosensory cortex displayed higher intensity in mutant mice than that in wild-type mice, suggesting more complex axon projection induced by the *Fgf13* mutation (*Figure 3E, F*). The axons also projected to additional cortical regions including secondary somatosensory cortex (S2) in mutant mice (*Figure 3E*), consistent with the increased number of axon branches in cultured neurons induced by the *Fgf13* mutation (*Figure 3C, D*). Additionally, the complexity of contralateral axon projection was also remained to be increased at P14 in the mutant mice (*Figure 3—figure supplement 1E, F*). Furthermore, we performed Golgi staining to evaluate the dendritic spine density in the pyramidal neurons from both S1 and CA1 hippocampal regions of mutant mice. The spine density of either basal or apical dendrites was decreased in *Fgf13* mutant mice (*Figure 3—figure supplement 1G*). Taken together, these data suggest that the point mutation of *Fgf13* increases the axon complexity, impairs the precise projection of axons in the cerebral cortex, and reduces the spine density of cortical and hippocampal neurons in adult mice, suggesting the disorganization of neural circuits in the brain.

## The 5'-UTR SNP reduces the ribosome association and translation efficiency of FGF13 mRNA

To further evaluate the specific role of this *FGF13* mutation in regulating protein translation, we investigated the global and FGF13 mRNA translation in the cortex of wildtype and mutant mice. The ribosome subunits (40S, 60S), monosome (80S) and polysome fractions were separated using the polysome profiling method. The polysome-to-monosome ratios of cortex in wildtype and mutant mice were similar in the polysome profiling chart, in which the amount of ribosomal RNAs in different fractions was detected by UV detector at 254 nm, suggesting that the global translation activity of mutant mice was not changed (*Figure 4A*). We further examined the *Fgf13* translation activity by

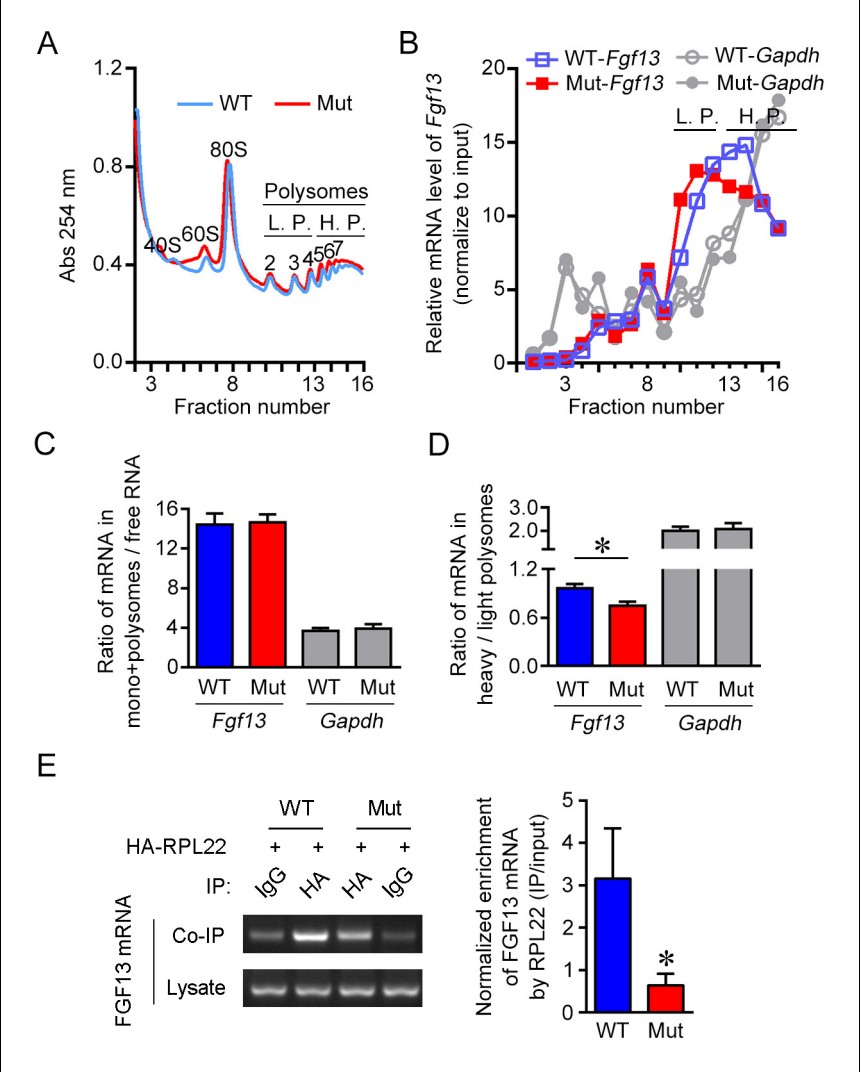

**Figure 4.** The 5'-untranslated region (5'-UTR) point mutation impairs association of FGF13 mRNA with ribosomes. (A) The overall mRNA levels in the brain of wildtype (WT) or mutant (Mut) mouse revealed by UV absorption profiles (254 nm). A total of 16 fractions were collected from the sucrose density gradients. Components of ribosome subunits (40S, 60S), monosome (80S) and polysomes are indicated in the curve. The light polysomes (L. P.) represent 2–4 polysomes, and heavy polysomes (H.P.) represent five and more polysomes. (B) Polysome profiles show the association level of FGF13 mRNA and GAPDH mRNA with ribosomes in the Mut mouse compared with the WT mouse. The distribution of FGF13 mRNA and GAPDH mRNA in different fractions after normalization to the input is shown here. The level of FGF13 mRNA in the heavier polysome fractions of Mut mouse was lower than that of WT mouse, while the level of GAPDH mRNA was not altered. (C, D) The ratio of FGF13 or GAPDH mRNA in monosomes plus polysomes to free RNA in Mut mice was not changed (WT: n = 6 mouse brains; Mut: n = 6 mouse brains; p=0.9372 for FGF13 and p=0.8182 for GAPDH by Mann–Whitney U test). The ratio of FGF13 mRNA, but not GAPDH mRNA, in the heavy polysome versus that in the light polysome of Mut mice was lower than that of WT mice (WT: n = 6 mouse brains; Mut: n = 6 mouse brains; p=0.0260 [*] for FGF13 and p=0.9372 for GAPDH by Mann–Whitney U test). (E) The 5'-UTR point mutation impaired the binding of FGF13 mRNA with translating complex. RNA immunoprecipitation (RIP) assay was used to examine the ribosome-binding mRNAs, in which FGF13 mRNA enriched in ribosomes was detected through IP with antibody against HA in HEK293 cells expressing HA-RPL22 with WT or Mut mouse FGF13 isoform 2. Representative PCR results are shown in the left panel, and quantitative real-time PCR shows a reduced enrichment of Mut FGF13 mRNAs with HA-RPL22 (n = 5, p=0.0355). Data are represented as the mean ± SEM (C–E). *p<0.05 versus WT by Student's t-test (D, E).

The online version of this article includes the following source data and figure supplement(s) for figure 4:

**Source data 1.** Source data for *Figure 4*.

*Figure 4 continued on next page*

*Figure 4 continued*

**Figure supplement 1.** The association of FGF13 mRNA with heavy polysomes is decreased in *Fgf13* 5′-untranslated region (5′UTR) mutant mice.

**Figure supplement 1—source data 1.** Source data for *Figure 4—figure supplement 1*.

detecting the levels of FGF13 mRNA in different fractions and used *Gapdh* as a control (***Figure 4B***, ***Figure 4—figure supplement 1A***). The ratio of FGF13 or GAPDH mRNA in ribosomes (monosome plus polysomes) divided by free RNA was not changed, implying that total translating mRNAs were not changed in mutant mice (***Figure 4C***). Importantly, the ratio of FGF13 mRNA in the heavy fractions (≥5 polysomes) versus that in the light fractions (2–4 polysomes) was lower in mutant mice compared to that in wildtype mice, while the ratio of GAPDH mRNA remained unchanged in mutant mice (***Figure 4D***). Additionally, the semi-quantitative PCR following the polysome profiling (***Tahmasebi et al., 2014***) also detected that FGF13 mRNA showed less association with heavy polysomes in the cortex of mutant mice, but the total mRNA level of FGF13 was not altered (***Figure 4—figure supplement 1B, C***). As a control, the distribution patterns of GAPDH mRNA were similar in the cortex of wildtype and mutant mice (***Figure 4—figure supplement 1B, C***). Thus, the efficiency of *Fgf13* translation is reduced in mutant mice.

We further examined the association of FGF13 mRNA with translating ribosomes through the immunoprecipitation of ribosome large-subunit protein RPL22 that was tagged by HA at its N-terminus (HA-RPL22) (***Sanz et al., 2009***). The mouse FGF13 mRNA binding to ribosomes could be enriched through the immunoprecipitation with HA antibody (***Figure 4E***). Quantitative real-time PCR showed that the amount of HA-immunoprecipitated FGF13 mRNA was decreased in HEK293 cells expressing the mouse *Fgf13* 5′-UTR mutant compared to the wildtype (***Figure 4E***). Taken together, the 5′-UTR point mutation impairs the association of FGF13 mRNA with translating ribosomes, therefore reduces the translation efficiency.

## The *FGF13* 5′-UTR SNP disrupts its binding with PTBP1/2

We next explored the underlying mechanism for the impaired translation of FGF13 mRNA in cells expressing the mutant 5′-UTR of *FGF13*. Although the IRES could stimulate the cap-independent translation on RNA, we found that the 5′-UTR of *FGF13* had no IRES activity with the dual-luciferase analysis (***Figure 5—figure supplement 1A***). On the other hand, the secondary structure of 5′-UTR RNA may influence the ribosome scanning and the recognition of AUG initiation codon (***Chatterjee and Pal, 2009***). However, the point mutation of *FGF13* did not alter the secondary structure of the core 5′-UTR mRNA by online prediction (***Figure 5—figure supplement 1B***). The potential uORF could be defined by an initiation codon in the frame with a termination codon located upstream of the main AUG of RNAs, which may reduce the efficiency of translation initiation in the main downstream ORF (***Barbosa et al., 2013***). None of the uORFs was found in the FGF13 transcripts using the website prediction.

Then, we examined the RNA-binding proteins differentially bind to wildtype and mutant RNA sequence. A short RNA derived from *FGF13* 5′-UTR carrying triple repeats of 17 bp wildtype or mutant sequence was synthesized and labeled with biotin in the N-terminus (see Materials and methods and ***Supplementary file 5***). The RNA pull-down assay was performed to enrich the RNA-binding proteins recognizing *FGF13* 5′-UTR in HEK293 cells (***Figure 5A***). Liquid chromatography tandem-mass spectrometry (LC-MS/MS) detected the top 10 enriched proteins pull-downed by the synthesized RNA derived from wildtype *FGF13* 5′-UTR (5′-biotin-CUUCCGUCUUCUGAGCG-CUUCCGUC UUCUGAGCG-CUUCCGUCUUCUGAGCG-3′) compared to the negative control with the blank RNA in HEK293 cells (***Figure 5B***). Notably, PTBP1 displayed the highest enrichment and a clearly higher affinity with the wildtype sequence than the mutant one (5′-biotin-CUUCCGUCUUGUGAGCG-C UUCCGUCUUGUGAGCG-CUUCCGUCUUGUGAGCG-3′), while some other proteins including MATR3 had equivalent binding affinity with both wildtype and mutant sequences (***Figure 5B***). PTBP1 expressed in most tissues and cell types is an intensely studied RNA-binding protein involved in pre-mRNA splicing and other steps of mRNA metabolism including polyadenylation, mRNA stability, and initiation of protein translation (***Romanelli et al., 2013***). Immunoblotting also confirmed the binding of wildtype sequence of synthesized RNA with PTBP1 and a decrease in the binding affinity

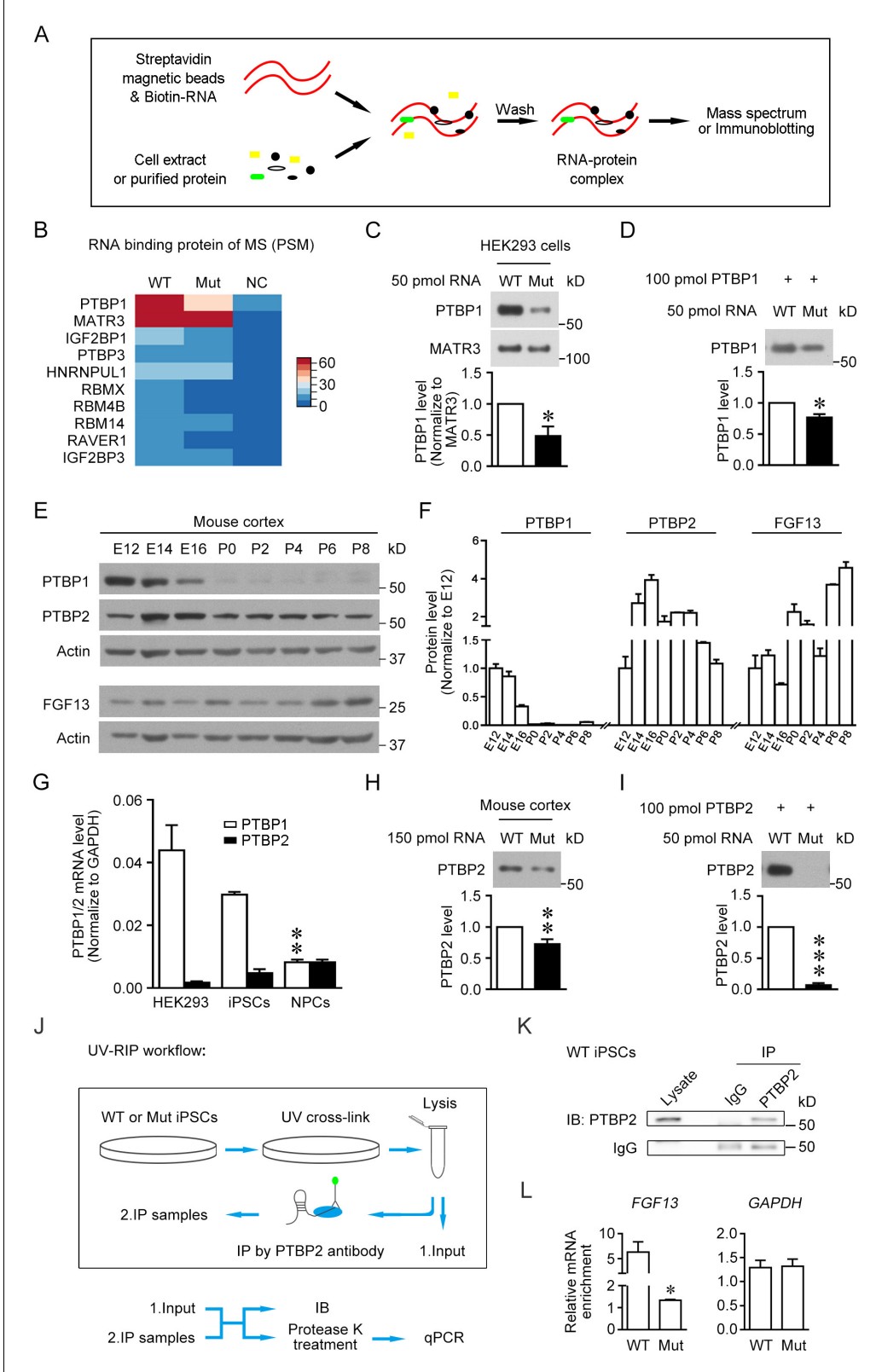

**Figure 5.** The 5′-untranslated region (5′-UTR) point mutation impairs the binding of PTBP1/2 with *FGF13* 5′-UTR. (**A**) Schematic graph shows the procedure of RNA pull-down assay. Biotin-labeled synthesized RNA was perfused with cell extracts or purified protein and enriched in streptavidin magnetic beads that were further eluted for mass spectrum or immunoblotting detection. (**B**) Top 10 ranked liquid chromatography tandem-mass spectrometry (LC-MS/MS) results of pull-downed proteins with synthesized RNAs of wildtype (WT) *FGF13* 5′-UTR and mutant (Mut) *FGF13* 5′-UTR are

*Figure 5 continued on next page*

*Figure 5 continued*

shown in heatmap. PTBP1 and MATR3 had fundamental enrichment with the WT RNA, but the Mut RNA selectively abolished its binding with PTBP1. NC indicates negative control without RNA in pull-down experiment. (**C**) Immunoblotting confirmed the decreased binding affinity of synthesized Mut RNA with endogenous PTBP1 but not MATR3 in HEK293 cells (n = 3, p=0.0399). (**D**) Immunoblotting shows the direct binding between purified PTBP1 (100 pmol) and synthesized *FGF13* RNA (50 pmol). The mutation decreased its binding level with PTBP1 (n = 3, p=0.0107). (**E, F**) Immunoblotting shows the change of FGF13, PTBP1, and PTBP2 in the mouse cerebral cortex at various developmental stages. FGF13 (isoform 2) was increased persistently from E14 to P8. PTBP1 was continuously decreased from E12 to P0, and largely absent at P0–P8. The PTBP2 expression reached the peak at E14–E16, then decreased and kept at certain levels at P0–P8 (n = 2–4 mice at each stage). (**G**) The mRNA level of PTBP1 and its neuronal paralog PTBP2 were examined in HEK293 cells, human induced pluripotent stem cells (iPSCs), and induced neural progenitor cells (NPCs). PTBP2 was specifically enriched in neural cells in which PTBP1 was decreased (n = 3 for iPSCs and n = 4 for HEK293 and NPCs; NPCs versus iPSCs: p=0.0034 [**] for PTBP1, p>0.9999 for PTBP2, by two-way ANOVA followed with Bonferroni's multiple comparisons test). (**H**) Immunoblotting showed that the homologous point mutation in *Fgf13* 5'-UTR hampered the binding of mouse PTBP2 with synthesized Mut RNA of the *Fgf13* 5'-UTR. Biotin-labeled synthesized RNAs were incubated with the mouse cerebral cortex extract to pull-down binding proteins (n = 10, p=0.0027). (**I**) Immunoblotting shows the decreased binding affinity between purified PTBP2 (100 pmol) and synthesized Mut *FGF13* RNA (50 pmol). The point mutation almost totally abolished their interaction (n = 3, p=0.0006). (**J**) Flowchart of ultraviolet -RNA immunoprecipitation (UV-RIP) experiments in WT or Mut iPSCs. (**K**) Immunoblotting showed that PTBP2 was immunoprecipitated by PTBP2 antibody. (**L**) Quantitative real-time PCR shows that the abundant *FGF13* in PTBP2 immunoprecipitates of WT iPSCs was largely decreased compared with that of Mut iPSCs (n = 3, p=0.0204). *GAPDH* was not apparently enriched and used as control (n = 3, p=0.4581). The mRNA enrichment was defined as the ratio of immunoprecipitated sample to input sample and normalized to corresponding IgG group. Data are represented as mean ± SEM (**C, D, F, G–I, L**). *p<0.05, ***p<0.01, and ***p<0.001 versus WT by Student's *t*-test (**C, D, H, I, L**).

The online version of this article includes the following source data and figure supplement(s) for figure 5:

**Source data 1.** Source data for *Figure 5*.

**Figure supplement 1.** The *FGF13* 5'-untranslated region (5'-UTR) single-nucleotide polymorphism (SNP) does not induce the alteration of internal ribosome entry site (IRES) entry and mRNA structure.

**Figure supplement 1—source data 1.** Source data for *Figure 5—figure supplement 1*.

of PTBP1 with the mutant sequence of synthesized RNA in HEK293 cells (*Figure 5C*). Moreover, the purified PTBP1 protein also showed a lower binding affinity with the mutant sequence of *FGF13* 5'-UTR than that with the wildtype one (*Figure 5D*, *Figure 5—figure supplement 1C*). Thus, the point mutation attenuates the binding of PTBP1 with the *FGF13* 5'-UTR.

PTBP2 (also named as nPTB or brPTB) has been reported to be the neural paralog of PTBP1 and is increased with the neuronal differentiation (*Keppetipola et al., 2012*). Reanalysis of the published data of single-cell RNA sequencing from human fetal prefrontal cortex showed that both PTBP1 and PTBP2 mRNAs were abundant in human NPCs, but PTBP2 mRNA was more enriched than PTBP1 mRNA in neurons (*Figure 5—figure supplement 1D*; *Zhong et al., 2018*). To verify the expression pattern of PTBPs in the nervous system, PTBP1 and PTBP2 were examined in mouse cerebral cortex at various developmental stages (*Figure 5E, F*). The PTBP1 expression was decreased persistently from E12 to E16 and largely absent at P0–P8. The PTBP2 expression was increased at E14–E16, but decreased at P0 to a certain level that remained at least to P8 (*Figure 5F*). The decrease of PTBP1 and increase of PTBP2 are presumed to link with neuronal maturation (*Boutz et al., 2007*; *Zheng et al., 2012*). The PTBP2 mRNA was also abundant in the induced NPCs compared with that in HEK293 cells and human iPSCs, while PTBP1 was largely decreased in NPCs (*Figure 5G*). These results imply that PTBP2 plays an important role during brain development.

Therefore, we detected the binding of PTBP2 with the *Fgf13* 5'-UTR in mice. Notably, the synthesized mutation-carrying RNA of *Fgf13* (5'-biotin-CUUCCUUCUCGUGGACG-CUUCCUUCUCG UGGACG-CUUCCUUCUCGUGGACG-3') displayed a lower binding affinity with PTBP2 in cell extract from mouse cortex compared to that of the wildtype one (5'-biotin-CUUCCUUCUCC UGGACG-CUUCCUUCUCCUGGACG-CUUCCUUCUCCUGGACG-3') (*Figure 5H*). In addition, we also purified human PTBP2 to check the direct binding with the triple repeats of 17 bp core sequence of the wildtype *FGF13* 5'-UTR (*Figure 5I*). Consistently, immunoblotting showed that the purified PTBP2 could bind to the wildtype sequence of synthesized RNA, and the point mutation largely hampered this binding affinity (*Figure 5I*, *Figure 5—figure supplement 1C*). Furthermore, we detected the binding of PTBP2 with endogenous *FGF13* 5'-UTR in human iPSCs. Ultraviolet ray was used to crosslink RNA and its binding proteins (*Figure 5J*), after which PTBP2 antibody was used in this RNA immunoprecipitation (RIP) experiment to enrich FGF13 mRNA (*Figure 5J, K*). The highly enriched (approximately fivefolds) FGF13 mRNA by PTBP2 in wildtype iPSCs was largely

decreased in mutant iPSCs with *FGF13* 5′-UTR SNP (*Figure 5L*). Taken together, the 5′-UTR SNP attenuates the binding of *FGF13* 5′-UTR with both PTBP1 and PTBP2.

## Increasing PTBP2 binding to the 5′-UTR SNP of *Fgf13* rescues protein translation and neuronal development

Given that the 5′-UTR mutation of *FGF13* impaired its binding with PTBP1 and PTBP2, we further tested the importance of this mechanism leading to the defect of *FGF13* translation and neuronal development. Knockdown of highly expressed PTBP1 in HEK293 cells (*Figure 6A*) with small interference RNA (siRNA) indeed reduced the protein level of wildtype *FGF13* reporter, while the mRNA level was not reduced (*Figure 6A*), suggesting that PTBP1 is required for *FGF13* translation. Then, we examined whether PTBP2, expressed in mouse cerebral cortex at P0–P8 when PTBP1 was largely absent (*Figure 5E*), was required for the *Fgf13* translation in cortical neurons. PTBP2 was distributed in both nucleus and cytoplasm of neurons cultured from mouse cerebral cortex at P0 (*Figure 6—figure supplement 1*). Immunoblotting showed that the FGF13 level was decreased in cultured cortical neurons when PTBP2 was knocked down (*Figure 6B*). Thus, the *FGF13*/*Fgf13* translation requires PTBP1 and PTBP2.

We further tested whether the PTBP2/*FGF13* 5′-UTR interaction was essential for the *FGF13* translation. Firstly, we inserted the MS2-binding motif into *FGF13* 5′-UTR (MS2-FGF13) and fused MS2 coat protein with PTBP2 (MCP-PTBP2) to increase its association with FGF13 mRNA (*Morisaki et al., 2016*; *Figure 6C*). RIP experiments detected that the exogenous MS2-FGF13 mRNA exhibited an increased binding to MCP-PTBP2 in HEK293 cells (*Figure 6D*). At the same time, MCP-PTBP2 increased the protein level of FGF13 in HEK293 cells compared with PTBP2 alone (*Figure 6E*), suggesting that the enhanced binding of PTBP2 to *FGF13* 5′-UTR is able to increase the protein translation. Then, we try to rescue the PTBP2 binding to endogenous mutant *Fgf13* 5′-UTR in neurons of mutant mice. Pumilio/fem-3 mRNA-binding factor (PUF) was reported to specifically recognize and bind to a fragment of 8-nucleotide RNA sequence to artificially increase the binding affinity between the RNA and the PUF-fused protein (*Dong et al., 2011*). According to the engineering principle (*Choudhury et al., 2012*), PTBP2 was fused with 'wildtype-recognition' PUF (wPUF-PTBP2, recognizing the wildtype RNA sequence UCUC**C**UGG) or with 'mutant-recognition' PUF (mPUF-PTBP2, recognizing the mutant RNA sequence UCUC**G**UGG), respectively (*Figure 6F*). Notably, PTBP2 in these two fusion proteins could normally bind to wildtype mouse *Fgf13* 5′-UTR, but had a lower binding affinity with mutant *Fgf13* 5′-UTR. Thus, wPUF-PTBP2 binds to wildtype *Fgf13* 5′-UTR either through wPUF or PTBP2, but both bindings are reduced in mutant *Fgf13* 5′-UTR; mPUF-PTBP2 could enhance the binding to mutant *Fgf13* 5′-UTR through its mPUF domain. The binding affinity was further verified by RNA pull-down assay. The level of mPUF-PTBP2 pull-downed by the synthesized mutant RNA was increased to the level similar to that by the synthesized wildtype RNA, while wPUF-PTBP2 could only partially bind with the synthesized mutation-carrying RNA (~65% of binding with the synthesized wildtype RNA) (*Figure 6G*), suggesting that the engineered mPUF-PTBP2 increases the binding of PTBP2 with mutant *Fgf13* 5′-UTR. Moreover, we tested whether the increase in the binding affinity between PTBP2 and mutant *Fgf13* 5′-UTR could enhance the *Fgf13* translation in mice. For efficient expression of wPUF-PTBP2 and mPUF-PTBP2 in cultured cortical neurons, we used lentivirus to infect neurons of *Fgf13* mutant mice (see Materials and methods). Immunoblotting showed that the expression of mPUF-PTBP2 increased the FGF13 protein level in the cortical neurons of mutant mice, although wPUF-PTBP2 had no effect on FGF13 expression (*Figure 6H*). Therefore, the enhanced binding of PTBP2 to mutant *Fgf13* 5′-UTR is able to rescue the impaired protein translation.

We finally examined whether abnormal neuronal development of mutant mice could be rescued by mPUF-PTBP2. Cortical neurons cultured from *Fgf13* mutant mice had decreased polarity and increased axon branching, while expressing mPUF-PTBP2 but not wPUF-PTBP2 could rescue these defects (*Figure 6I*), consistent with the finding that the expression of mPUF-PTBP2 rather than wPUF-PTBP2 increased the FGF13 protein level in cortical neurons of mutant mice (*Figure 6H*). These data support that enhancing PTBP2 binding to mutant *Fgf13* 5′-UTR rescues abnormal neuronal morphology in mutant mice.

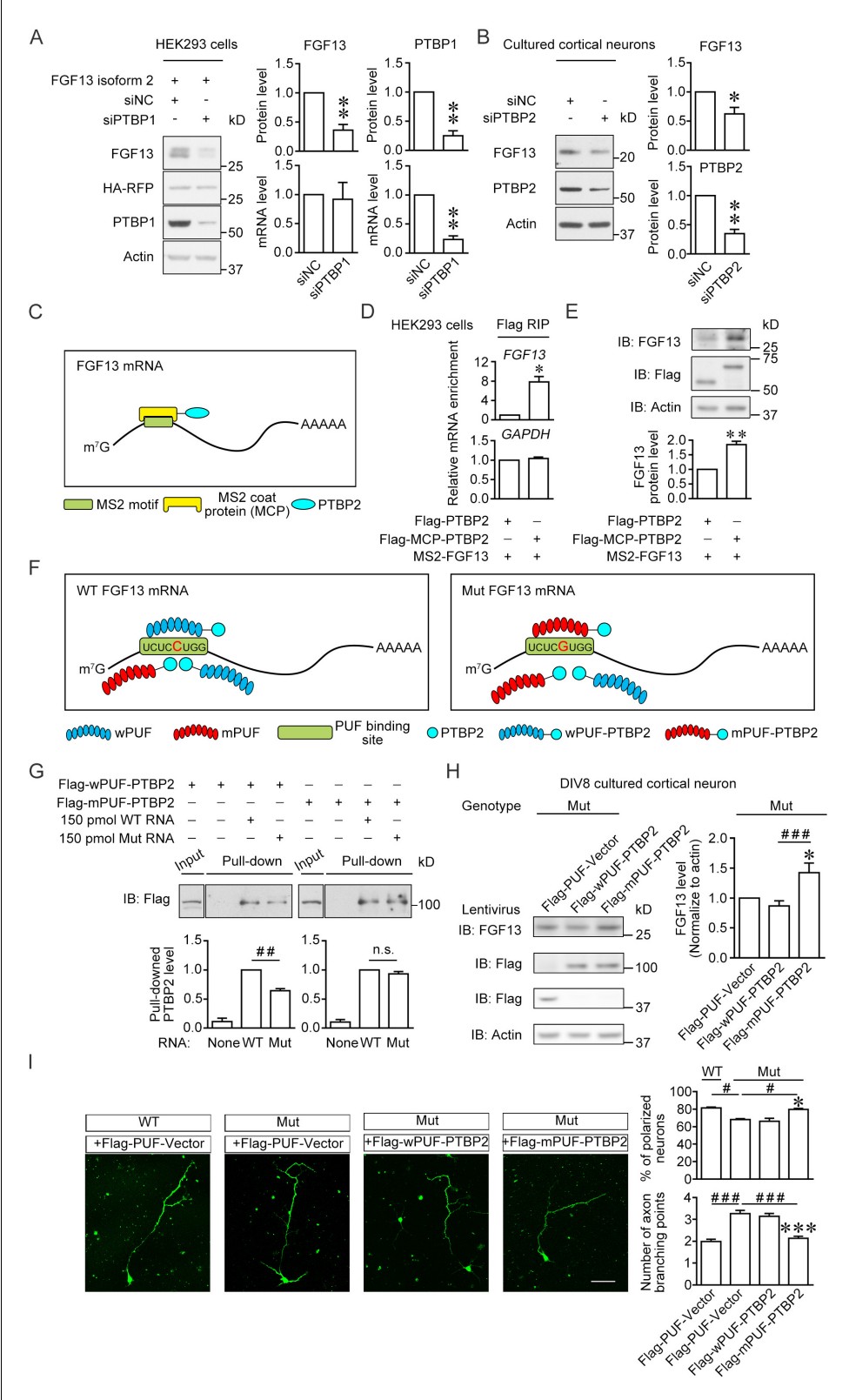

**Figure 6.** Enhancement of the PTBP2 binding with mutant (Mut) *Fgf13* 5′-untranslated region (5′-UTR) rescues the impairment of neuronal morphogenesis. (**A**) Immunoblotting showed that the PTBP1 knockdown reduced the translation of co-expressed human FGF13 isoform 2 in HEK293 cells (n = 4, p=0.0077 for FGF13, p=0.0032 for PTBP1). The mRNA level of FGF13 remained unchanged when PTBP1 was silenced (n = 3, p=0.8100 for FGF13, p=0.0066 for PTBP1). (**B**) Immunoblotting showed that the PTBP2 knockdown reduced the FGF13 (isoform 4) expression in cultured mouse

*Figure 6 continued on next page*

Figure 6 continued

neurons (n = 4, p=0.0425 for FGF13, p=0.0018 for PTBP2). Three different small interference RNAs (siRNAs) targeting PTBP2 were mixed to enhance the knockdown efficiency in neurons. (**C**) Schematic graph shows that MCP-fused PTBP2 is recruited to the MS2-inserted *FGF13* 5′-UTR. (**D**) Quantitative real-time PCR showed that exogenous MS2-FGF13 mRNA exhibited an increased binding to Flag-MCP-PTBP2, after RNA immunoprecipitation (RIP) by Flag beads in HEK293 cells (n = 3, p=0.0262), while GAPDH mRNA was not enriched (n = 3, p=0.3807). (**E**) Immunoblotting showed that MS2-PTBP2 overexpression increased the protein level of FGF13 in HEK293 cells compared to PTBP2 (n = 4, p=0.0054). (**F**) Schematic graph shows that engineered PTBP2 protein with PUF scaffold recognizes 8 nt RNA sequence and is recruited to the *Fgf13* 5′-UTR. Due to the difference in designed recognition RNA sequence (the mutation site of *Fgf13* is highlighted in red color), wPUF recognizes and binds to wildtype (WT) 5′-UTR, while mPUF binds to mutant (Mut) 5′-UTR. However, the PTBP2 within fused proteins normally binds to WT *Fgf13* 5′-UTR but has a lower binding affinity with Mut *Fgf13* 5′-UTR. (**G**) Immunoblotting showed that wPUF-PTBP2 with Flag tag at the N-terminus had lower binding level with synthesized Mut RNA compared to synthesized WT RNA (n = 3, p=0.0099), but mPUF-PTBP2 reversed the binding difference between synthesized WT and Mut RNAs (n = 3, p=0.2386). Proteins of input were simultaneously immunoblotted. $^{##}$p<0.01 and not significant (n.s.) versus indicated group by Student's *t*-test. (**H**) The engineered PTBP2 with PUF scaffold could increase the FGF13 expression in cultured neurons of Mut mice. Lentivirus expressing PUF-Vector, wPUF-PTBP2, and mPUF-PTBP2 were used to transfect cortical neurons cultured from Mut mice. Immunoblotting showed that the FGF13 expression was increased by mPUF-PTBP2, but not by wPUF-PTBP2 (n = 8). p=0.0357 (*) versus Flag-PUF-Vector and p=0.0008 ($^{###}$) versus indicated group by Student's *t*-test. (**I**) Neurons from the Mut mice exhibited a lower ratio of polarized neurons with single axon and an increase in axon branching compared to that of WT mice. Introducing wPUF-PTBP2 did not alter the aberrant neuronal morphogenesis, but mPUF-PTBP2 could increase the ratio of polarization and decrease the axon branching. Neurons were immunostained with axon marker SMI-312. n = 100, 92, 78, and 95 neurons for four groups, respectively, from three experiments. For polarization ratio: p=0.0500 (*) and p=0.0500 (*) versus Flag-wPUF-PTBP2, while p=0.0500 ($^{#}$) versus indicated group by Mann–Whitney U test. For axon branching: p=1.67E-10 (***) and p=1.88E-08 (***) versus Flag-wPUF-PTBP2, while p=3.65E-09 ($^{###}$) versus indicated group by Mann–Whitney U test. Scale bar, 50 µm. Data are represented as mean ± SEM (**A**, **B**, **D**, **E**, **G**–**I**). *p<0.05 and **p<0.01 by Student's *t*-test (**A**, **B**, **D** and **E**).

The online version of this article includes the following source data and figure supplement(s) for figure 6:

**Source data 1.** Source data for *Figure 6*.

**Figure supplement 1.** The subcellular distribution of PTBP2 in mouse cortical neurons at DIV4.

## Discussion

The present study identified a pathogenic 5′-UTR SNP of *FGF13* in Chinese Han children. In normal individuals, PTBP2 binds to the 5′-UTR of *FGF13* to promote the ribosome entry and translation of mRNAs in neurons (*Figure 7*). However, in ID patients, the C>G variation of *FGF13* 5′-UTR results in a decreased binding with PTBP2 and reduces the *FGF13* translation, thus leading to the impaired brain development and cognitive functions (*Figure 7*). Therefore, we find a specific 5′-UTR SNP that causes the defects in FGF13 protein synthesis and the ID in ID patients.

Accumulated evidence suggests that FGF13 may be involved in the pathogenesis of XLID (*Gecz et al., 1999*; *Goldfarb, 2005*). Early studies proposed that duplication in Xq26 may contribute to XLID (*Gecz et al., 1999*; *Solomon et al., 2002*; *Gécz et al., 2009*) due to a disruption of the coding or regulatory region of genes located at its boundaries or induction of a dosage effect of proteins encoded by genes located within the duplicated region. In a Börjeson–Forssman–Lehmann syndrome-like patient, *FGF13* was interrupted by a duplication breakpoint (*Gecz et al., 1999*). Furthermore, the impairment of learning and memory in FGF13-deficient mice provides a direct evidence for the possibility of FGF13 involved in XLID (*Wu et al., 2012*). In a genetic epilepsy family associated with cognitive impairment, a chromosome translocation disrupts *FGF13* and reduces the expression of FGF13, which are speculated to cause enhanced excitability within local circuits of hippocampus, thereby resulting in the clinical phenotype of epilepsy (*Puranam et al., 2015*). In the present study, a point mutation of *FGF13* 5′-UTR in a subpopulation of male patients with severe ID phenotypes further illuminates the role of FGF13 in XLID.

From the survey of 100 severe ID children in a Chinese Han population, the phenotypes of ID carrying the *FGF13* mutation almost exclusively occurred in males but not in female carriers, consistent with the principle of non-crisscross inheritance of XLID. The allele frequency regarding this point mutation is much higher in severe ID patients (~2%) than that in general population of various online database (~0.2%), revealing that the SNP of *FGF13* 5′-UTR is highly enriched in the ID population. In three families of the present study, all male carriers exhibited ID phenotypes. However, whether the allele frequency in general population (~0.2%) comes from unaffected female carriers or from reduced penetrance of male carriers is still unknown. The true penetrance of the SNP in general population could only be estimated based on data from more families of affected individuals. In the small ID subpopulation including three families, both exon capture-sequencing and whole-genome

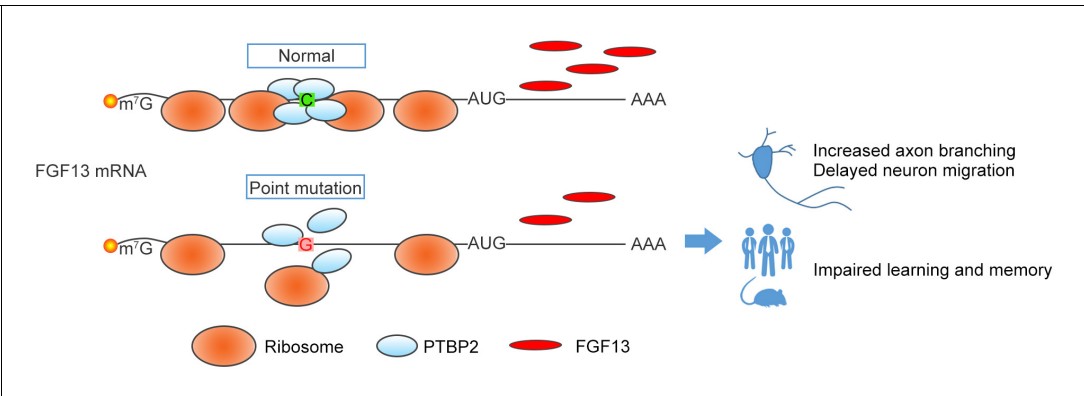

**Figure 7.** A proposed model of the *FGF13* 5'-untranslated region (5'-UTR) single-nucleotide polymorphism (SNP) in attenuating protein translation and impairing brain development. In normal situation, PTBP2 binds to the 5'-UTR of *FGF13* and recruits the ribosomes to initiate the protein translation. However, the C>G mutation in the *FGF13* 5'-UTR of intellectual disability (ID) patient impairs the PTBP2 binding, attenuates the ribosome association and translating complex recruitment, and therefore causes the decrease in FGF13 protein synthesis. During brain development, the FGF13 reduction results in increased axon branching and delayed neuronal migration, and finally leads to the cognitive impairment in both ID patients and corresponding mouse models.

sequencing were performed to prove that this point mutation of *FGF13* 5'-UTR was the exclusively dominant ID-causing variant due to the absence of other reported ones related to intellectual development. Thus, our study provides the first SNP identified in *FGF13* that contributes to XLID in Chinese Han population.

During brain development, cortical neurons migrate from the site of their last mitotic division toward final destination to generate neural circuits (*Ayala et al., 2007*; *Marín et al., 2010*; *Yuan, 2010*). Regulation of microtubule dynamics is crucial for a neuronal transition from the multipolar to bipolar morphology, which is a prerequisite for initiating the neuron migration (*Witte et al., 2008*; *Jaglin and Chelly, 2009*; *Li et al., 2012*; *Zhang et al., 2012*). MSPs are involved in this regulatory process (*Poulain and Sobel, 2010*). FGF13 is transiently expressed in the cerebral cortex from E14 until P14 with a peak level at P7 (*Wu et al., 2012*). FGF13 is another recently identified MSP, and the deficiency of FGF13 disrupts the MT stabilization, thus leading to axon/leading process branching and delayed neuronal migration in both cerebral cortex and the CA1 region of hippocampus in mice. Learning and memory were impaired in FGF13-deficient mice, although the positioning of neurons in the cerebral cortex and hippocampus of adult mutant mice became similar to that in normal mice (*Wu et al., 2012*). In the present study, mice with the point mutation of *Fgf13* 5'-UTR showed the defects of neuronal polarization, axon branching, and neuronal migration similar to that of *Fgf13* knockout mice, consistent with the finding that the point mutation attenuated the FGF13 expression in mouse brain. Notably, the impaired spine formation in the hippocampus suggests that the aberrant neural circuit establishment in adult *Fgf13* mutant mice may involve in the behavioral deficits. In addition to the function of FGF13 in neuronal development, FGF13 was also reported to bind to and regulate the function of voltage-gated sodium channels at the Ranvier nodes along the myelinated nerve fibers (*Wittmack et al., 2004*), at the axon initial segment of hippocampal neurons (*Goetz et al., 2009*), and in DRG neurons (*Yang et al., 2017*). Further analysis of electrophysiological recordings to examine the efficiency of synaptic transmission in mutant mice could help to understand the mechanisms correlated with the defects of learning and memory behavior.

Mutations influencing the coding region of genes could alter the protein sequence and result in the dysfunctions in many human diseases. However, mutations in the noncoding region of genes are involved in various regulating mechanisms and relatively difficult to study. Studies using the exome sequencing to screen variations in the gene landscapes hardly give information on the noncoding

region of genome that have important roles in regulating gene transcription, translation, stability, and many other aspects (*Chatterjee and Pal, 2009*; *Barrett et al., 2012*). Genes involved in numerous important cellular functions, such as fertilization, development, and cell cycle, are regulated at the level of translation. Certain changes in the UTR of genes are associated with various diseases, indicating the critical function of noncoding regions (*Signori et al., 2001*). Moreover, mutations located on 5′-UTR are related to several genetic diseases (*Kondo et al., 1998*; *Tassone et al., 2001*; *Scheper et al., 2007*; *Wang et al., 2007*) and involved in different ways to influence the gene translation (*Chatterjee and Pal, 2009*). The 5′-UTR length, GC content, and thermal stability, as well as its secondary structures, multiple uORFs, and IRES could influence overall translation rate. Furthermore, the 5′-UTR has numerous binding sites for proteins that may repress or promote translation (*Chatterjee and Pal, 2009*; *Barrett et al., 2012*). For example, a G>C mutation in *BRCA1* 5′-UTR was identified in a highly aggressive sporadic breast cancer and resulted in a defective initiation of BRCA1 protein translation (*Signori et al., 2001*).

In the present study, we provide several lines of evidence showing that the SNP in the *FGF13* 5′-UTR of ID patients causes a reduction of *FGF13* translation by attenuated binding with RNA-binding proteins PTBP1/2. First, the point mutation in *FGF13* 5′-UTR decreased the expression of FGF13 in HEK293 cells, patient iPSCs, and *Fgf13* mutant mice. Second, the polysome profiling assay and examination of mRNA binding on translating ribosomes showed that the ribosome entry of FGF13 mRNAs was reduced, which may cause a decreased efficiency of *Fgf13* translation in the mutant mice. Third, the RNA pull-down assay detected that the 5′-UTR SNP attenuated the binding of *FGF13* 5′-UTR with both PTBP1 and PTBP2. Finally, an increase of the binding between PTBP2 and *FGF13* 5′-UTR with MS2/MCP system or mutant *Fgf13* 5′-UTR with mPUF-PTBP2 rescued the translation of FGF13. Specific increase of the binding affinity between RNA and related PUF-fused PTBP2, rather than direct overexpression of PTBP2, may bypass potential side effects due to the important biological roles of PTBP2 in neuronal development (*Hu et al., 2018*). Obviously, our study provides another case for the importance of noncoding SNPs in the dysfunction of protein synthesis and the pathogenicity of ID. PTBP1/2 usually play major roles in regulating pre-mRNA splicing by binding to pyrimidine-rich sequences. They could also affect miRNA function and cytoplasmic translation (*Sawicka et al., 2008*; *Kafasla et al., 2012*). Our study provides further evidence for the cytoplasmic function of PTBPs in FGF13 translation.

The next-generation sequencing of various disease cases could produce numerous data across whole human exome or genome (*Exome Aggregation Consortium et al., 2016*). To find the pathological mutations related to ID among these massive data, we need a fast approach to screen these mutations. First effort is usually made to screen out pathogenic variants from remaining background ones (*Gilissen et al., 2014*), and the second step is to verify the pathogenic functions and mechanisms. The generation of patient-derived iPSCs from easily harvested renal epithelial cells (*Zhou et al., 2012*) and CRISPR/Cas9 gene editing may provide convenient tools to study the functions of numerous gene mutations and thus accelerate the studies of ID genomic spectrum (*Takahashi et al., 2007*; *Jinek et al., 2012*; *Mali et al., 2013*; *Ran et al., 2013*). Subsequently, the induced NPCs and functional neurons from iPSCs further facilitate the study of the functions and mechanisms of neural diseases in human (*Chambers et al., 2009*; *Shi et al., 2012*; *Wapinski et al., 2013*; *Zhang et al., 2013*; *Sakaguchi et al., 2015*; *Tao and Zhang, 2016*). The mouse models inspired by mutations from human diseases could perform standard examinations from neurons and neural circuits to behaviors; therefore, new technologies to generate the mutation-introduced mice (*Zhong et al., 2015*) provide us with a rapid and efficient strategy to screen the pathogenic mutations from big scale of data, thus leading to the 'phenotype-omics' for each genetic variations. Recently, the semi-cloning technology was used to identify the pathogenic combinations of genetic variants in mouse model for a complex disease Müllerian anomalies (*Wang et al., 2020*). It would largely benefit the genetic etiological and pathological studies of ID and any other diseases. However, the development of human brain is more complicated, and some human neural diseases have more severe phenotypes than those in mice (*Gleeson et al., 1998*; *Kappeler et al., 2006*). The studies are still required to correlate the degree of changes at cell and mouse models with the patients suffering from human neural diseases.

# Materials and methods

## Key resources table

| Reagent type (species) or resource | Designation | Source or reference | Identifiers | Additional information |
|---|---|---|---|---|
| Genetic reagent (*Mus musculus*) male and female | *Fgf13* mutant | This paper | Home-made | Obtained by contacting the correspondence author |
| Antibody | Anti-FGF13 (goat polyclonal) | Santa Cruz | Cat# sc-16811 RRID:AB_2104044 | IB (1:500) |
| Antibody | Anti-FGF13 (rabbit polyclonal) | *Wu et al., 2012* | Home-made by immunization with FGF13B 182–192 aa | IB (1:5000) |
| Antibody | Anti-HA (mouse monoclonal) | Sigma | Cat# H3663 RRID:AB_262051 | IB (1:2000); IP (1 μg) |
| Antibody | Anti-Flag (mouse monoclonal) | Sigma | Cat# F3165 RRID:AB_259529 | IB (1:10,000) |
| Antibody | Anti-actin (mouse monoclonal) | Chemicon | Cat# MAB1501 RRID:AB_2223041 | IB (1:400,000) |
| Antibody | Anti-MATR3 (rabbit monoclonal) | Abcam | Cat# ab151739 RRID:AB_2885091 | IB (1:2000) |
| Antibody | Anti-PTBP1 (mouse monoclonal) | This paper | Home-made from ATCC Cat# CRL-2501 | Source: RRID:CVCL_L596; IB (1:2000) |
| Antibody | Anti-PTBP2 (mouse monoclonal) | Abcam | Cat# ab57619 RRID:AB_2284865 | IB (1:500); IP (4 μg) |
| Antibody | Anti-Lamin B1 (mouse monoclonal) | Proteintech | Cat# 66095-1-Ig RRID:AB_2721256 | IB (1:5000) |
| Antibody | Anti-SOX2 (rabbit polyclonal) | Stem Cell Technologies | Cat# 60055 RRID:N/A | ICC (1:100) |
| Antibody | Anti-Tuj1 (mouse monoclonal) | Chemicon | Cat# CBL412 RRID:AB_11205398 | ICC (1:1000) |
| Antibody | Anti-Tuj1 (chicken polyclonal) | Abcam | Cat# ab107216 RRID:AB_10899689 | ICC (1:1000) |
| Antibody | Anti-GFP (chicken polyclonal) | Abcam | Cat# ab13970 RRID:AB_300798 | ICC/IHC (1:2000) |
| Antibody | Anti-SMI-312 (mouse monoclonal) | BioLegend | Cat# 837904 RRID:AB_2566782 | ICC (1:1000) |
| Antibody | Anti-Cux1 (rabbit polyclonal) | Santa Cruz | Cat# sc-13024 RRID:AB_2261231 | IHC (1:1000) |
| Recombinant DNA reagent | Plasmids | This paper | | *Supplementary file 6* |
| Sequence-based reagent | Oligonucleotides, primers | This paper | | *Supplementary file 5* |

IB: immunoblotting; IP: immunoprecipitation; IHC: immunohistochemistry; ICC: immunocytochemistry.

A complete list of plasmids, primers, and antibodies used in this study is provided in *Supplementary file 5* and *Supplementary file 6*.

## Animals

Experiments were performed according to the recommendations in the Guide for the Care and Use of Laboratory Animals of the National Institutes of Health (8th Edition, 2010) and were approved by the Committee of Use of Laboratory Animals and Common facility, Institute of Neuroscience, Chinese Academy of Sciences (CAS). C57BL/6J mice (RRID:IMSR_JAX:000664) were raised together with the littermates in the pathogen-free environment, and their health status was routinely checked. No more than five mice were housed in one cage. Food and water were provided ad libitum. Mice were maintained in a 12 hr light/dark cycle at 22–26°C. Experiments were conducted during the light

phase of the cycle. The 2- to 3-month-old male littermate mice and age-matched C57 mice were used for behavioral experiments, and in utero electroporation experiments were performed on pregnant female mice at E14.5. Littermate genetically modified pup mice were used for primary cortical culture (P0, male and female), immunohistochemistry analysis (P0, P7, and P14, male and female), and polysome profiling assay (P4, male).

To generate the genetically modified mice carrying point mutation, both Cas9 mRNA and sgRNA targeting *Fgf13* 5′-UTR were in vitro transcribed using MESSAGE mMACHINE T7 ULTRA kit (ThermoFisher Scientific, AM1345) and MEGAshortscript T7 kit (ThermoFisher Scientific, AM1354), respectively, according to the manufacturer's instructions. RNAs were then purified with MEGAclear Transcription Clean-Up Kit (ThermoFisher Scientific, AM1908) and stored at −80℃. Zygotes were collected and injected with Cas9 mRNA, sgRNA, and ssOligo donor at one-cell stage. Successful two-cell stage embryos were then transferred into each oviduct of pseudo-pregnant ICR female mice at 0.5 day postcoitum (dpc). Surviving pups were raised by lactating mouse mothers. The designed sgRNA and DNA donor sequences are listed in *Supplementary file 5*. Mice carrying the *Fgf13* mutation were identified by PCR screening with primers listed in *Supplementary file 5*. C57BL/6J mice were purchased from Shanghai Laboratory Animal Center, CAS (Shanghai, China).

## Patient subjects

The ethical approval for this study was approved by the ethical institutional review board of Xu-Hui Central Hospital (Shanghai, China). Written informed consent for genetic studies was obtained from all participants. All study procedures and protocols in this study comply with the Institute of Neuroscience and State Key Laboratory of Neuroscience, CAS Center for Excellence in Brain Science and Intelligence Technology (Shanghai, China). A total of 272 children with ID from Shanghai specific education school were recruited to our study in the Shanghai Clinical Research Center of CAS based at the Xu-Hui Central Hospital. Inclusion criteria included the diagnosis of male or female ID patients by a clinical geneticist, and the disease severity was categorized using DSM–IV classification. Functional assessments of the ID were detected by the test of Wechsler Intelligence Scale for Children. 100 ID individuals who were diagnosed with severe ID (IQ < 40) and conclusive diagnostic reports were included in this study.

## DNA sequencing and variant calling

The DNA samples were sequenced at Shanghai Biotechnology Corporation. Genomic DNAs were extracted from the blood samples using DNeasy Blood and Tissue Kit (Qiagen). Firstly, Sanger sequencing of *FGF13* was performed in 100 severe ID cases and a novel SNP was found in *FGF13* of 3 male ID children. A custom SureSelect DNA target enrichment panel was designed to cover 262 reported ID-related genes (from screening of database and literatures, see *Supplementary file 1*) and the target region capture-sequencing was performed on Illumina 2500 platform in all three male ID subjects and their parents to assess the overall landscape of ID-related SNPs. Finally, to assess the overall variants in whole genomes of the three ID cases, we performed whole-genome sequencing on the Illumina HiSeq 2500 platform to obtain all SNPs, In-Dels, and CNVs located at both coding and noncoding regions. Sequence reads were aligned to the human reference genome GRCh37/hg19 using the Burrows-Wheeler Alignment software (BWA, version 0.5.9) and further processed to call variants following the Genome Analysis Toolkit (GATK) Best Practices workflow (*McKenna et al., 2010*). Target region capture-sequencing covered 100% of target regions and had a mean depth in target region over 170×. Whole-genome sequencing had 91.03%, 91.03%, and 91.07% coverage of whole-genome region with depths of 23.48, 33.96, and 25.04, respectively, in three ID individuals. SNPs and In-Dels were called using GATK Unified Genotyper software followed by annotation using ANNOVAR software and 1000 Genomes, dbSNP, ExAC, and clinVar databases. CNVs were called using CNVnator_v0.3 software and annotated using ANNOVAR software and Decipher database.

## Plasmid construction

The 5′-UTR and CDS from cDNAs of human and mouse FGF13 isoform 2 and isoform 4 were cloned into pcDNA3.1/myc-his(-) for the expression of different FGF13 isoforms. The full-length cDNA of HA-fused RFP driven by a CAG promoter was subcloned into the plasmids as the transfection control. The wildtype or mutant 5′-UTR of *FGF13* was cloned into psiCHECK-2 dual-luciferase expression

vectors, in which they were inserted before the Renilla (Rluc) open reading frame (ORF) of psi-CHECK-2 vector. The cDNA of human RPL22 were cloned into the pCDNA3 vector with a HA tag inserted into the N-terminus using the KOD-plus mutagenesis kit (Toyobo). The wildtype or mutant 5′-UTR of *FGF13* was cloned into pRIF-Rluc-ECMV IRES-Firefly luciferase vector in which they replaced the ECMV IRES fragment before the Firefly ORF. The sgRNA sequences of human and mouse FGF13 were inserted into the PX459 plasmid for iPSCs and mouse genome editing. The cDNAs of human and mouse PTBP1 and PTBP2 were cloned into the pCDNA3 vector and the pFast-Bac1 vector, and a Flag tag was inserted into the N-terminus of PTBPs. The MS2 motif was inserted at the −32 position of *FGF13* 5′-UTR in the pCDNA3.1-hFGF13-TV2 vector. The MCP, which recognizes MS2 motif, was inserted into pcDNA3-Flag-hPTBP2 between Flag and PTBP2 to obtain fused proteins. The PUF protein recognizing specific eight sequential nucleotides of RNA (*Choudhury et al., 2012*; *Wang et al., 2013*) was used to enhance the binding of PTBP2 with FGF13 mRNAs. The coding sequences of customized PUF proteins, which recognize the RNA sequence of UCUCCUGG or UCUCGUGG, were subcloned into pcDNA3-Flag-mPTBP2 between Flag and mouse PTBP2 to obtain fused proteins. All primers and oligonucleotides are listed in *Supplementary file 5*.

## Cell culture, transfection, and lentivirus infection

HEK293 cells (RRID:CVCL_0045) were from the Cell Bank of the Chinese Academy of Sciences (Shanghai, China), authenticated by STR profiling and routinely tested for mycoplasma contamination before use. HEK293 cells were cultured in Minimum Essential Medium (MEM, Invitrogen) containing 10% fetal bovine serum. The cells were transfected with 1–4 μg plasmids using Lipofectamine 2000 reagent (Invitrogen). In siRNA repression experiments, 50 nM siRNA targeting *PTBP1* (GenePharma Company) was transfected using Lipofectamine RNAiMAX reagent (Invitrogen). The cells were used for the following experiments 24–48 hr after transfection.

Cortical neurons were cultured from rodent pups at P0 as described previously (*Beaudoin et al., 2012*). Briefly, the cerebral cortex tissues from P0 mice were carefully isolated and digested in papain (10 U/mL, Worthington Bio Corp) and DNase (0.1 mg/mL, Sigma) for 20 min at 37°C. After gentle trituration, dissociated neurons were resuspended and counted, and electroporated with 2 μg plasmid using the Amaxa Nucleofector II System (Lonza Amaxa, Germany), or transfected with 50 nM siRNA (mixture of three siRNAs targeting different region of *Ptbp2*) using Lipofectamine RNAi-MAX reagent. Neurons were cultured in Neurobasal medium supplemented with 2% of B27 (Invitrogen) and 200 mM of L-glutamine (Invitrogen), and were seeded in culture plates previously coated with poly-D-lysine (0.01 mg/mL, Sigma) at the concentration of 500,000 cells/mL. Lentivirus was generated using the ViraSafe lentiviral packaging system (Cell Biolabs). Lentiviral vectors were constructed from pCDH-CMV-EF1-puro vector and expressing Flag-PUF-Vector, Flag-wPUF-wPTBP2, and Flag-wPUF-mPTBP2. Neurons were infected at the first day of cultured and the medium was totally replaced 24 hr after infection, and were used for following experiments after cultured for 1 week. Mutant mice for analysis included both male hemizygous and female homozygous mice.

## Immunoblotting

The tissues dissected from the mouse brains and the cultured cells were lysed in the Gross buffer (20 mM Tris-HCl [pH 7.5], 150 mM NaCl, 5 mM MgCl$_2$, 1% NP-40, 1 mM PMSF, 10 mg/mL aprotinin, 1 mg/mL pepstatin A, and 1 mg/mL leupeptin). The equivalent denatured samples were subjected to SDS-PAGE, transferred, and probed with antibodies against FGF13 (1:500, Goat, RRID:AB_2104044), FGF13 (1:5000, Rabbit; home-made; *Wu et al., 2012*), Flag (1:10,000, Mouse, RRID:AB_259529), HA (1:2000, Mouse, RRID:AB_262051), Actin (1:400,000, Mouse, RRID:AB_2223041), MATR3 (1:2000, Rabbit, RRID:AB_2885091), PTBP1 (1:2000, Mouse, home-made from ATCC Cat# CRL-2501), PTBP2 (1:500, Mouse, RRID:AB_2284865), Lamin B1 (1:5000, Mouse, RRID:AB_2721256), and visualized with the enhanced chemiluminescence. We also stained the SDS-PAGE gels with Coomassie brilliant blue R-250 (Sangon Biotech) to detect the purified proteins and visualized by a scanner. The immunoreactive bands were quantified from 3 to 5 independent experiments using the ImageJ software, and the results were normalized against the corresponding actin or HA-RFP band as loading control.

## Quantitative real-time PCR and semi-quantitative PCR

Total mRNAs of cultured cells or the cerebral cortex and the hippocampus of mice were extracted with TRIzol reagent (Invitrogen). SuperScript II reverse transcriptase (Invitrogen) was used for reverse transcription to produce the complementary DNAs. Real-time PCR was performed with the SYBR Premix Ex Taq (Takara) reagents and the ABI Prism 7500 apparatus. Quantification was performed through normalizing with endogenous GAPDH or self-made standard curve (diluting one starter sample in gradient), or normalizing with indicated group. Primer pairs are listed in *Supplementary file 5*. Semi-quantitative PCR was also performed with the SYBR Premix Ex Taq (Takara) reagents, and optimal number of cycles was tested and determined beforehand so that amplification of each primer was in the exponential range (*Chen et al., 1999*).

## Human iPSC induction

Human iPSCs were derived from the exfoliated renal epithelial cells in the urine, according to the standard procedure (*Zhou et al., 2012*). WT iPSCs were purchased from South China Stem Cell Bank ( *Xue et al., 2013*). In brief, the urine was collected and centrifuged for the isolation of urinary cells that were then incubated at 37°C for approximately 2 weeks for proliferation. When the urinary cells became dense enough for passaging, they were subcultured into new dishes and infected with SKOM sendai virus that supplied four Yamanaka factors including SOX2, KLF4, OCT4, and c-MYC. After another round of infection and further culture, the infected urinary cells with iPSC-like morphology were trypsinized and plated on the mouse embryonic fibroblast (MEF) feeder cells. After several days, the human embryonic stem cell (hESC)-like colonies were mechanically picked and plated onto feeder cells to proliferate.

## NPCs and neuronal differentiation

Control iPSCs and iPSCs in which the *FGF13* SNP was introduced through Cas9-editing were used in our studies to be differentiated into NPCs (*Marchetto et al., 2010*) and then differentiated into neurons (*Zhang et al., 2016*). The iPSC colonies were plated on the Matrigel-coated (BD Biosciences) plates and maintained in the mTESR medium (Stem Cell Technologies) for 5 days, then the medium was changed to the Gibco Dulbecco's Modified Eagle Medium: Nutrient Mixture F-12 (DMEM/F-12 medium, Gibco) supplemented with $0.5\times$ N2 (Gibco), $0.5\times$ B27 supplements (Gibco), 1 µM dorsomorphin (Tocris), and 1 µM SB431542 (Selleck). After 2 days, the colonies were removed and cultured in suspension as embryoid bodies for 8 days using $0.5\times$ N2 medium. Embryoid bodies were then plated on the dishes coated with 10 µg/mL polyornithine (Sigma) and 5 µg/mL laminin (Gibco), and maintained in the DMEM/F12 medium supplemented with $0.5\times$ N2, $0.5\times$ B27 supplements, and 20 ng/mL basic fibroblast growth factor (bFGF). Rosettes emerging after 3 days or 4 days were manually selected, gently dissociated with accutase (Stemcell), and replated. The NPCs were maintained in the DMEM/F12 medium with N2, B27, and bFGF. The NPCs were differentiated into neurons in the absence of bFGF for at least 3 weeks and with 10 µM ROCK inhibitor (Y-27632; Calbiochem) treatment during the first 24 hr.

## Immunohistochemistry/immunocytochemistry

For immunohistochemistry, mice were perfused with 4% paraformaldehyde (PFA) and fixed overnight (P0-adult mice) in 4% PFA at 4°C. Fixed brains were cryo-protected for 1–2 days in 20–30% sucrose/phosphate buffered saline (PBS) at 4°C and mounted in OCT compound and sectioned coronally (50 µm) with a cryostat (Leica). Brain sections were first blocked with 5% bovine serum albumin and 0.1% Triton X-100 for 1 hr at room temperature and then incubated with primary antibodies against Cux1 (1:1000, Rabbit, RRID:AB_2261231) or GFP (1:2000, Chicken, RRID:AB_300798) overnight at 4°C. For immunocytochemistry, cultured neurons or cells were fixed with 4% PFA and 0.2% picric acid, and then incubated with primary antibodies against SOX2 (1:100, Rabbit, Stem Cell Technologies Cat# 60055), Tuj1 (1:1000, Mouse, RRID:AB_11205398; 1:1000, Chicken, RRID:AB_10899689), GFP (1:2000, Chicken, RRID:AB_300798), or SMI-312 (1:1000, Mouse, RRID:AB_2566782) overnight at 4°C. Then, the sections and cells were incubated with correspondent FITC-, Cy3-, or/ and Cy5-conjugated secondary antibodies for 45 min at 37°C, and then counterstained with DAPI (1:2000, Sigma) for 15 min at room temperature before coverslipping. The images were acquired using a Leica SP8 confocal microscope (Leica, Germany) and used for the morphological analysis

using the ImageJ software (RRID:SCR_003070). For quantitative analysis of neuronal radial migration and axonal intensity, three brain sections from each of 3–4 mice were collected for each group. The total cell number of migrating neurons was manually counted with the ImageJ cell counter tool. The fluorescence intensity along the radial axis of cortical wall in the ipsilateral and contralateral cortex was measured and normalized by the axon intensity in the ipsilateral layer V, according to a previous study (*Dan et al., 2018*). For the analysis of axon branches in cultured neurons, 30–50 neurons from each of 3–5 independent experiments were collected for each group. A process longer than 20 µm was defined as a branch, and the number of total axon branches was counted.

## Protein purification

The coding regions of PTBP1 and PTBP2 were cloned into the pFastBac1 vectors, respectively. The Bac-to-Bac baculovirus expression system (Invitrogen) was used to generate recombinant baculovirus. P3 viruses were used to infect 100 mL sf9 cells ($2$–$3 \times 10^6$/ mL), and cells were collected after 72 hr. Cells were lysed by gentle sonication in the lysis buffer (50 mM $Na_2HPO_4$, pH 8.0, 300 mM NaCl, 2 mM $MgCl_2$, 5% glycerol, 1 mM PMSF), and the lysate was centrifuged for 30 min at 35,000 g. Anti-Flag M2 affinity gel (Sigma) were used to bind protein with the supernatant. Binding protein was dissolved in the elute buffer (0.2 mg/mL 3X Flag peptides, APExBio). After concentration, proteins were dialyzed in PBS and stored at $-80℃$.

## Upstream ORF and RNA secondary structure prediction

The uORFs are sequences defined by an initiation codon in a frame with a termination codon located upstream of the main AUG of RNAs. The NCBI website 'ORFfinder' (https://www.ncbi.nlm.nih.gov/orffinder/) was used to search for all ORFs in the RNA sequence of *FGF13*, including main ORF and uORFs. The 41 bp (20 bp upstream and downstream of the mutant site) RNA sequence from the *FGF13* 5′-UTR was sent to predict the secondary structure by the M-fold web server (http://unafold.rna.albany.edu/?q=mfold/RNA-Folding-Form) to determine whether the computed minimum energy folding was changed.

## Dual-luciferase reporter assay

The dual-luciferase reporter assay system (Promega) was conducted according to its instructions with a minor modification, and the GloMax 20/20 Luminometer (Promega) was used in the manual luciferase activity measurement. The *FGF13* 5′-UTR or the control sequence was inserted into the psiCHECK-2 vector or pRIF vectors, which were then transfected into HEK293T cells. The cells were collected with the passive lysis buffer (supplied in kit, 100 µL/well) 48 hr later. After incubation, 20 µL supernatant was mixed with 20 µL Luciferase Assay Reagent II for the measurement of Firefly luciferase activity by the luminometer, then 20 µL Stop and Glo reagent was added for the measurement of Renilla luciferase activity.

## In utero electroporation

In utero electroporation was performed as described previously (*Saito and Nakatsuji, 2001*) with minor modifications. Briefly, the uteruses of mice were exposed at gestation day 14.5 and 1–2 µg plasmids with 0.5 µL Fast Green (0.05%, Sigma) were injected into the lateral ventricle of each embryo. Plasmids were mixed with 0.5 µg/µL pCAG-YFP to label infected neurons. Five electrical pulses (30 V, 50 ms, 1 s interval) were applied across the uterine wall to embryos using ECM-830 BTX square wave electroporator (VWR International). The uterine horns were then replaced in the abdominal cavity, and the abdomen wall and skin were sutured. The mouse brain was performed for immunohistochemistry at different developmental stages (P0, P7, and P14). Mutant mice for analysis included both male hemizygous and female homozygous mice.

## RNA-protein pull-down assay

Pierce magnetic RNA-protein pull-down kit (Thermo Scientific) was used according to its manual. Briefly, we designed 51-nucleotide sequence that was triple repeats of 17 nucleotides covering the SNP site (the SNP was the 11th nucleotide in the 17-nucleotide sequence) of 5′-UTR of human (5′-biotin-CUUCCGUCUUC(G)UGAGCG-CUUCCGUCUUC(G)UGAGCG-CUUCCGUCUU C(G)UGAGCG-3′) and mouse (5′-biotin-CUUCCUUCUCC(G)UGGACG-CUUCCUUCUC C(G)UGGACG-CUUCCUUC

UCC(G)UGGACG-3′) FGF13 mRNA (C for wildtype and G for mutant sequence). The biotin-labeled sequence at 5′ end was commercially constructed (Nan-Jing Genscript Company, China). The 50 pmol (human) or 150 pmol (mouse) RNA was used per reaction, and was incubated with 50 µL Streptavidin magnetic beads (Thermo Scientific) at the room temperature. The RNA-bound beads equilibrated in the protein-RNA binding buffer were then incubated with the protein or cell lysate and separated on a magnetic stand. The samples were then eluted with 95% formamide and 80 mM NaOAc for the detection of mass spectrometry or immunoblotting.

## Polysome profiling assay

Polysome profiling analysis was carried out as previously described (*Gandin et al., 2014*). Briefly, cortical tissues of P4 male mice were dissected and homogenized with tissue lysis buffer (100 mM KCl, 5 mM $MgCl_2$, 10 mM HEPES [pH 7.4], 100 µg/mL cycloheximide, 1X protease inhibitor cocktail [EDTA-free], 100 units/mL RNase inhibitor, 25 units/mL Turbo DNase I, 2 mM DTT, 0.5% Triton X-100%, and 0.5% sodium deoxycholate). Debris was removed by centrifugation at 12,000 rpm for 10 min at 4℃, and supernatants were loaded onto 10 mL continuous 10–50% sucrose gradients (100 mM KCl, 5 mM $MgCl_2$, 10 mM HEPES [pH 7.4], 100 µg/mL cycloheximide, 1X protease inhibitor cocktail [EDTA-free], 100 units/mL RNase inhibitor) and centrifuged at 35,000 rpm for 2.5 hr at 4℃ in an SW41 rotor (Beckman). Gradients were fractionated by Brandel fractionation system (Brandel), and the optical density (OD) at 254 nm was continuously recorded using an ISCO UA-6 ultraviolet detector (Teledyne ISCO). The in vitro transcribed YFP mRNA (5 ng) was added into each collected fraction as spike-in RNA for the following normalization. Total RNA from each fraction was isolated using TRIzol (Invitrogen) and used for further analysis.

## RIP and UV crosslinking RIP

RIP was used for detecting the interaction of RNA-binding proteins with RNAs. Translating complex was immunoprecipitated by ribosome large-subunit protein RPL22 that was tagged by HA at N-terminus (HA-RPL22) as previously described (*Sanz et al., 2009*). HA-RPL22 was transfected into HEK293 cells together with the plasmids of wildtype or mutant mouse FGF13 isoform 2. MS2-FGF13 isoform 2 was transfected into HEK293 cells together with the plasmid of Flag-PTBP2 or Flag-MCP-PTBP2. UV crosslinking RIP was performed as previously described (*Sei and Conrad, 2014*). Briefly, UV irradiation (250 mJ/cm$^2$) was used to covalently crosslink RNA to protein in human iPSCs within ice-cold PBS for 1 min. Cells were lysed in polysome buffer (50 mM Tris, pH 7.5, 100 mM KCl, 12 mM $MgCl_2$, 1% Nonidet P-40, 1 mM DTT, 200 units/mL RNase inhibitor, 100 µg/mL cycloheximide, 1X protease inhibitor cocktail [EDTA-free]), then the lysate was centrifuged and 5% of the supernatant was taken for input sample. For immunoprecipitations, the supernatant was precipitated with 1 µg of anti-HA antibody (Sigma, RRID:AB_262051), or mouse IgG (Santa Cruz), or 4 µg of anti-PTBP2 antibody (Abcam, RRID:AB_2284865) at 4℃ overnight and afterward 50 µL protein G beads (Roche) at 4℃ for 2 hr, or with 30 µL of Anti-Flag M2 affinity gel (Sigma) at 4℃ for 2 hr. The immunoprecipitated sample was then washed three times for 5 min in high salt buffer (50 mM Tris, pH 7.5, 300 mM KCl, 12 mM $MgCl_2$, 1% Nonidet P-40, 1 mM DTT, 100 µg/mL cycloheximide). Total RNA was prepared using TRIzol (Invitrogen) and used for further analysis. For UV-crosslinking RIP, the RNA from input sample or beads was treated with protease K (0.5 mg/mL) at 37℃ for 1.5 hr, before being prepared with TRIzol (Invitrogen) and used for further analysis. The enrichment of mRNA transcript was defined as the ratio of mRNA in immunoprecipitated sample to input sample and was normalized to correspondence IgG group.

## Behavior tests

Adult male mice (8–10-week-old, the transgenic mice were the second-generation offspring of the Cas9-editing founder mice) were performed for the following behavior tests. Mice were adapted to be familiar with handling and experimenter's odor for five successive days before tests. Wildtype mice included both littermates with wildtype genotype and C57 mice of the same age (raised in the same environment of mutant mice). The experimenter was blind to the genotypes and ages of the mice.

## Morris water maze test

Standard procedure of Morris water maze test was used (*Vorhees and Williams, 2006*). Mice were trained to find the visible 10-cm-diameter platform in 120-cm-diameter tank with one trail for the first day. They were then trained to find the hidden platform for six consecutive days and performed with four trails from different entries per training day. In each trial, mice were allowed to swim until they found the hidden platform when starting from different, random locations around the perimeter of the tank. Then, mice were allowed to sit on the platform for 15 s before being picked up. Wild-type mice included three littermates with wildtype genotype and four C57 mice of the same age. Only mice that engaged in a goal-directed search were used for analysis; therefore, mice that spent more than half of each trail floating, engaged in thigmotaxis (defined as remaining with 4.0 cm of the perimeter of the tank), or made no attempt to escape from the water tank were not included (*Lione et al., 1999*). Two mutant mice, one wildtype littermate mouse and six C57 mice, met the exclusion criteria and were excluded from the study. During the probe trial day, the platform was removed from the tank. The escape latency and the time spent in each quadrant were recorded for 30 s by a video camera.

## Fear conditioning test

For contextual and cued memory tests, two rounds of training were performed, followed by 30 s rest after the training. Each round contained a 2 min exposure of mice to the conditioning box (context), followed by a cued tone (30 s, 300 Hz, and 90 dB sound) and a foot shock (2 s, 0.75 mA constant current) (*Wehner and Radcliffe, 2004*). The short-time contextual memory test was performed 3 hr later by re-exposing the mice for 6 min to the conditioning context (*Guan et al., 2009*). The cued memory test was performed 24 hr later by exposing the mice for 3 min to a novel context, followed by an additional 3 min exposure to a cued tone. Wildtype mice included 10 C57 mice of the same age.

## Tail suspension test

Automated tail suspension apparatus (TAILSUSP-1N96, MED Associates) was used to measure the immobility of suspended mice. Before testing, the mouse tail was wrapped by adhesive tape in a constant position three quarters of the distance from the tail base. The mice were suspended by passing the suspension hook through a metal chain, and the total time of immobility was analyzed automatically (*Porsolt et al., 2001*). Wildtype mice included 10 C57 mice of the same age.

## Rotarod test

Locomotor coordination and balance were measured by placing mice on an accelerating, 3 cm diameter and rotating drum for three trials with a minimum 15 min interval between trials. The rotarod started at 5 rpm and increased to 40 rpm over a 5 min period. The mean latency to fall over three trials was the dependent measure. Mice from each group were pretrained for adaptation while the fall latency of the following three trials was recorded. Wildtype mice included 10 C57 mice of the same age.

## Open-field test

Exploratory locomotor activity in a 30 min period was measured in an open field (45 × 45 cm) by a Digiscan apparatus (Accuscan Electronics). The total distance of horizontal moving during the whole procedure was recorded. Wildtype mice included 10 C57 mice of the same age.

## Novel object recognition test

Male adult mice were placed in an open field (24 × 24 cm) to get familiar with two identical objects for 10 min. To test for object recognition after the training-to-test interval (1 hr), one of the objects was then replaced with a novel object and the animals were exposed in the apparatus again for 5 min to measure the time spent on exploring each of the objects (*Bevins and Besheer, 2006*). Wildtype mice included 10 C57 mice of the same age.

### Three-chamber test

The social interaction test was performed as previously described (*Silverman et al., 2010*). In brief, a mouse was placed in the central chamber of a clear Plexiglas box divided into three interconnected chambers and was given the choice to interact with either an empty wire cup (located in one side chamber) or a similar wire cup with an unfamiliar mouse inside (located in the opposite chamber). Time sniffing each cup was measured. Wildtype mice included 10 C57 mice of the same age.

### Light-dark box test

The light-dark box consists of a clear side (light side, 27 × 27 cm) and a fully opaque side (dark side, 18 × 27 cm) separated by a partition with a small opening (12 × 5 cm). The light compartment was brightly illuminated with a 60 W light bulb (550–850 lux). At the beginning of the test, the mouse was placed in the dark compartment and allowed to freely explore both compartments for 6 min. Ambulation distance, latency to the first entry, and time spent in the light compartments were recorded. Wildtype mice included 10 C57 mice of the same age.

### Quantification and statistical analysis

The data are presented as mean ± SEM. Exact sample number (n) and p values are indicated in figure legends or result section. Two groups were compared by nonparametric Mann–Whitney U test or Student's paired or unpaired t-test (one- or two-tailed) according to whether they could pass normality test. Comparisons among multiple groups with two factors were performed by a two-way ANOVA with Bonferroni's post-hoc test for multiple comparisons. Statistical analysis was performed using Prism 7.0 (GraphPad Software, RRID:SCR_002798). All statistical significances were set as *$p<0.05$, **$p<0.01$, and ***$p<0.001$.

## Acknowledgements

We thank Drs. Xiang Yu, Ligang Wu, and Hong Cheng for comments on this project. We thank Drs. Min Zhang, Zhenning Zhou, and Haiyan Wu (Molecular and Cellular Biology Core Facility at Institute of Neuroscience, CAS) for their help in induction and maintenance of iPSCs and NPCs. We thank Xiaoyi Liao (School of Life Science and Technology, Shanghai Tech University) for her help in the maintenance of HEK293 cells. We thank Dr. Yanan Cao (Shanghai Clinical Center for Endocrine and Metabolic Diseases, Ruijin Hospital, Shanghai) for providing the allele frequency of the SNP in ChinaMAP database. We thank the Core Facility at Shanghai Research Center for Brain Science and Brain-Inspired Intelligence. This work was supported by the National Natural Science Foundation of China (31630033 and 31991194), the Science and Technology Commission of Shanghai Municipality (18JC1420301), the Strategic Priority Research Program (B) of CAS (XDPB1005 and XDB39000000), and the key research program of frontier sciences, CAS (QYZDY-SSW-SMC007).

## Additional information

### Funding

| Funder | Grant reference number | Author |
| --- | --- | --- |
| National Natural Science Foundation of China | 31630033 | Xu Zhang |
| National Natural Science Foundation of China | 31991194 | Lan Bao |
| Science and Technology Commission of Shanghai Municipality | 18JC1420301 | Xu Zhang |
| Chinese Academy of Sciences | XDPB1005 | Xu Zhang |
| Chinese Academy of Sciences | XDB39000000 | Xu Zhang |
| Chinese Academy of Sciences | QYZDY-SSW-SMC007 | Xu Zhang |

The funders had no role in study design, data collection and interpretation, or the decision to submit the work for publication.

## Author contributions

Xingyu Pan, Conceptualization, Resources, Data curation, Software, Formal analysis, Validation, Investigation, Visualization, Methodology, Writing - original draft, Project administration, Writing - review and editing, X-YP carried out animal behavior tests, primary neuron culture, in utero electroporation, biochemistry and immunocyto/histochemistry. He also designed the project and wrote the manuscript; Jingrong Zhao, Conceptualization, Resources, Data curation, Software, Formal analysis, Validation, Investigation, Visualization, Methodology, Writing - original draft, Writing - review and editing, J-RZ performed molecular cloning, iPSCs induction, CRISPR/Cas9 editing, RNA pull-down and LC-MS/MS analysis. He also designed the project and wrote the manuscript; Zhiying Zhou, Conceptualization, Resources, Project administration, Z-YZ recruited ID patients; Jijun Chen, Resources, Methodology, J-JC contributed to induction and maintenance of iPSCs and NPCs; Zhenxing Yang, Conceptualization, Resources, Software, Project administration, Z-XY did sequencing and analysis of patient genome data; Yuxuan Wu, Resources, Methodology, Y-XW generated mutant mice; Meizhu Bai, Resources, Methodology, M-ZB generated mutant mice; Yang Jiao, Resources, Methodology, YJ contributed to protein purification and molecular cloning; Yun Yang, Resources, Formal analysis, Methodology, YY contributed to polysome profiling assay; Xuye Hu, Conceptualization, Resources, Supervision, Funding acquisition, Investigation, Methodology, Writing - original draft, Project administration, Writing - review and editing, X-YH contributed to induction and maintenance of iPSCs and NPCs; Tianling Cheng, Resources, Methodology, T-LC contributed to induction and maintenance of iPSCs and NPCs; Qianyun Lu, Resources, Methodology, Q-YL provided cDNAs of PUF proteins; Bin Wang, Resources, Methodology, BW contributed to protein purification and molecular cloning; Chang-Lin Li, Resources, Supervision, Methodology, C-LL contributed to data analysis of single-cell RNA-seq; Ying-Jin Lu, Resources, Supervision, Methodology, Y-JL contributed to immunoblotting; Lei Diao, Resources, Methodology, LD contributed to protein purification and molecular cloning; Yan-Qing Zhong, Resources, Methodology; Jing Pan, Resources, Methodology, JP contributed to immunoblotting; Jianmin Zhu, Resources, J-MZ recruited ID patients; Hua-Sheng Xiao, Resources, Software, Supervision, Methodology, H-SX did sequencing and analysis of patient genome data; Zi-Long Qiu, Resources, Formal analysis, Supervision, Methodology, Z-LQ chose the ID-related genes and contributed to induction and maintenance of iPSCs and NPCs; Jinsong Li, Resources, Supervision, Methodology, J-SL generated mutant mice; Zefeng Wang, Conceptualization, Resources, Supervision, Methodology, Z-FW contributed to polysome profiling assay and provided cDNAs of PUF proteins; Jingyi Hui, Conceptualization, Resources, Formal analysis, Supervision, Methodology, J-YH contributed to design of RNA pull-down and mutant mice; Lan Bao, Conceptualization, Resources, Supervision, Funding acquisition, Validation, Investigation, Writing - original draft, Project administration, Writing - review and editing, LB designed the project and wrote the manuscript; Xu Zhang, Conceptualization, Resources, Supervision, Funding acquisition, Investigation, Writing - original draft, Project administration, Writing - review and editing, XZ designed the project and wrote the manuscript

## Author ORCIDs

Xingyu Pan ⓘ https://orcid.org/0000-0001-9588-5830
Jinsong Li ⓘ https://orcid.org/0000-0003-3456-662X
Lan Bao ⓘ https://orcid.org/0000-0001-9269-9565
Xu Zhang ⓘ https://orcid.org/0000-0002-9617-6590

## Ethics

Human subjects: The ethical approval for this study was approved by the ethical institutional review board of Xu-Hui Central Hospital (Shanghai, China), with approval identifier number: 2012-32-addition-1; 2014-29; 2014-29-addition-1; 2017-035. Written informed consent for genetic studies was obtained from all participants. All study procedures and protocols in this study comply with Institute of Neuroscience and State Key Laboratory of Neuroscience, CAS Center for Excellence in Brain

Science and Intelligence Technology (Shanghai, China). Genomic DNAs of subjects were extracted from the blood samples and used for further sequencing.

Animal experimentation: This study was performed in strict accordance with the recommendations in the Guide for the Care and Use of Laboratory Animals of the National Institutes of Health. All of the animals were handled according to approved institutional animal care and use committee (IACUC) protocols (#NA-008-2016) of the Institute of Neuroscience, Chinese Academy of Sciences. The protocol was approved by the Committee of Use of Laboratory Animals and Common facility, Institute of Neuroscience, the Chinese Academy of Sciences. All surgery was performed under anesthesia with pentobarbital sodium, and every effort was made to minimize suffering.

## Decision letter and Author response
Decision letter https://doi.org/10.7554/eLife.63021.sa1
Author response https://doi.org/10.7554/eLife.63021.sa2

## Additional files

### Supplementary files
• Supplementary file 1. List of 262 reported intellectual disability-related genes from database and literature screening.

• Supplementary file 2. List of single-nucleotide polymorphisms and insertions and deletions of three families from exon-capture-sequencing.

• Supplementary file 3. List of single-nucleotide polymorphisms, insertions and deletions, and copy number variations of three children from whole-genome sequencing.

• Supplementary file 4. Clinical records of three intellectual disability individuals.

• Supplementary file 5. Primers and oligonucleotides used for real-time PCR, Cas9 targeting, sequencing, plasmid construction, and biotin probe.

• Supplementary file 6. Antibodies, reagents, cell lines, plasmid constructs, software, and other resources used in this study.

• Transparent reporting form

### Data availability
Data of whole genome sequencing reported in this study have been deposited in the CNSA (https://db.cngb.org/cnsa/) of China National GeneBank (CNGB) with the accession number of CNP0000742. All data generated or analysed during this study are included in the manuscript and supporting files. Source data for all manuscript figures are presented and uploaded within the submission.

The following dataset was generated:

| Author(s) | Year | Dataset title | Dataset URL | Database and Identifier |
|---|---|---|---|---|
| Pan XY | 2019 | Pathological mechanism study of intellectual disability | https://db.cngb.org/search/project/CNP0000742/ | China National GeneBank, CNP0000742 |

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
