## [Decision Letter]

Thank you for submitting your article "5'-UTR SNP of FGF13 causes translational defect and intellectual disability" for consideration by *eLife*. Your article has been reviewed by 3 peer reviewers, one of whom is a member of our Board of Reviewing Editors, and the evaluation has been overseen by Huda Zoghbi as the Senior Editor. The following individuals involved in review of your submission have agreed to reveal their identity: Fernanda Laezza (Reviewer #2); Gene W Yeo (Reviewer #3).

The reviewers have discussed the reviews with one another and the Reviewing Editor has drafted this decision to help you prepare a revised submission.

Summary:

This paper presents translational work aimed at connecting a SNP in the 5' UTR of FGF13 associated with intellectual disabilities with a surprising molecular mechanism involving the binding of a factor PTBP2 (usually associated with splicing) to enhance translational efficiency. The authors present a substantial amount of data including the analysis of protein production in vitro, in patient-derived iPSCs and in an ad hoc animal model carrying the mutation. While there were many strengths to the study, reviewers had concerns about the depth of analysis to support the novel molecular mechanism. The shift in the polysome profiles of the mRNA was modest and requires replicates to be convincing, the tethering reporter with PUF proteins was complicated by the endogenous proteins in the cell and would be clearer with a simpler experiment, and it was felt that more support was needed to establish interactions between PTBP2 and the SNP in the iPSCs.

Essential revisions:

1. Provide replicates for Figure 4B.

2. Consider a more simple tethering assay allowing for simpler interpretation and a more convincing conclusion.

3. Perform RIP type experiments in iPSCs to establish interaction between SNP and endogenous PTBP2.

4, Provide better graphic representation of the cellular phenotyping data (i.e. additional images, higher magnification, additional parameters measured)

*Reviewer #1:*

In this work, Pan et al. seek to expand our understanding of protein expression and how this contributes to human disease. The authors focus on FGF13 because of recent advancements suggesting its involvement in neural development and identify a C≥G SNP in the 5'UTR of FGF13 which is found in 3 male ID patients and is therefore enriched in the ID patient population relative to the general population (~2% compared to ~0.1-0.3%). Based on whole-genome sequencing of the affected individuals, the authors posit that this SNP may be causal for ID. The rest of the work takes this SNP into cell culture and mouse models in order to understand how it contributes to the ID phenotype. The authors observe modest isoform-specific reductions in protein levels of FGF13 in SNP-harboring HEK293 cells, iPSCs from patients, iPSC-derived neurons, and in mouse neural tissue. Mice feature slight behavioral and memory impairments, and slight defects in neural migration, polarization, and branching. Polysome profiling and ribosome-IP experiments suggest that the SNP slightly decreases FGF13 mRNA association with translating ribosomes. Mass spectrometry identifies PTBP1 and PTBP2 as proteins which preferentially associate with the WT 5'UTR of FGF13 over the SNP-containing 5'UTR. The authors show that decreasing PTBP expression also decreases FGF13 expression, and a tethering assay is attempted to show that recruitment of PTBP2 to the FGF13 5'UTR drives expression. Together, this combination of data leads the authors to propose a model whereby the C≥G SNP disrupts PTBP2 binding to the FGF13 5'UTR, leading to decreased FGF13 protein expression, disrupted neural development, and ultimately the ID phenotype.

While this work presents an intriguing model for regulation of FGF13, there are several key concerns that limit enthusiasm for publication at this stage.

1) The effect sizes throughout the cell culture and mouse model experiments are small. Slight decreases are observed in FGF13 protein expression and in neuronal migration and branching, and the mouse behavioral and memory impairments appear to be small and unenlightening due to the random assortment of positive and negative behavioral assays. These modest effect sizes stand in stark contrast to the severe ID phenotypes observed in the affected patients (IQ < 40, significant deficiencies in adaptive behaviors). This is concerning and calls to question whether the C≥G SNP is actually causal for the ID phenotype, or whether the mouse is an appropriate model to study the FGF13 5'UTR despite demonstrated conservation at the SNP site. A discussion of ID penetrance for humans harboring this SNP is warranted since ~0.1-0.3% of the population appears to carry this SNP.

2) The evidence for translational regulation is very modest – the small redistribution of FGF13 mRNA on the polysome gradient to a lighter fraction in Figure 4 seems unlikely to report on the ~25% decrease in protein expression reported in Figure 2D. This assay can be a useful one for looking at translational regulation but a greater shift would be more compelling (see Figure 2C in Tahmasebi et al., Cell Stem Cell (2014) for an example of a more compelling shift).

3) The authors model is predicated on a role of the PTBPs as enhancers of mRNA translation, whereas prior evidence suggests the major role of PTBPs to be in splicing. While this would be a novel and interesting role for PTBPs, the authors do not adequately discuss the known functions of PTBPs or canonical PTBP binding motifs, and do little experimentally to convince the reader of this novel function (e.g. subcellular localization). Finally, the experimental setup and interpretations of the tethering assay (Figure 6) are convoluted and provide no support for their model. Tethering is a valuable tool as it obviates the need for a well-defined binding site, but that is the way to utilize it – recruitment of PTBP2 to an orthogonal site such as an MS2 binding motif would be a cleaner method to show that PTBP recruitment is sufficient to drive translation of the FGF13 mRNA.

*Reviewer #2:*

This is a spectacular translational work by Pan et al. identifying a clinically relevant SNP of FGF13 in children affected by intellectual disabilities and demonstrating mechanistically the effect of the mutation in the gene expression and in the protein production in vitro, in patient-derived iPSC and in ad hoc animal model carrying the mutation. This work has a significant impact in the field and is supported by a strong premise built on rigorous data. This manuscript is publishable in its present form.

*Reviewer #3:*

The authors demonstrate that a C to G variant in the 5'UTR of the FGF13 gene results in decreased binding of an RNA binding protein PTBP2 which appears to affect translation was reduced. An increase of the binding between PTBP2 and mutant FGF13 5'UTR with a mutant PUF-PTBP2 rescue translation. The study also highlights a potentially interesting role for PTBP2 to enhance translation by 5'UTR binding, albeit it is not clear how that happens. The study has many positives, particularly the use of orthogonal systems (IPSC lines/neurons and mouse models) to evaluate the molecular, cellular and also behavioral impact of the 5'UTR mutation. The methods used are standard for the field, but nevertheless serve as a logical framework for tackling such a question. The prevalence of the mutation is low and while the mechanism described/revealed in this paper is interesting and highlights the relevance of non-coding mutations on translation control, particularly FGF13, in impaired learning and memory, it's difficult to conclude that that's the same as intellectual disability.

The writing in the paper is poor and there are many spelling errors and grammatical mistakes through the figures, figure legends and text. It really distracts from the science, which in contrast is actually expansion and interesting, using a range of orthogonal in vitro and in vivo approaches to identify the mechanism and consequences underlying a point mutation found in the 5'UTR of FGF13, suggested to impact protein translation of the gene.

1. Figure 1L. % of polarized neurons do not look convincing from the IHC images.

2. Figure 4 is not convincing (4a, 4b) as well.

3. PTBP2 knockout mice are available and phenotypes have been previously scored – can the authors discuss/make a comparison – is FGF13 isoform 4 levels lower in those mice?

4. The authors should have in the main figures EMSAs to show binding to the RNA and the mutant by PTBP2/PTBP1.

5. Can the authors show by CLIP/UV-RIP the binding of PTBP1/2 to the endogenous wildtype and mutant RNA?

---

## [Author Response]

Essential revisions:1. Provide replicates for Figure 4B.

According to the editor’s and reviewer’s comments, we performed further experiment to verify this result of Figure 4B. The semi-quantitative PCR following the polysome profiling from 3 wildtype and 3 mutant mice also detected that FGF13 mRNA showed less association with heavy polysomes in mutant mice. This result was shown as suggested in the format of that in Tahmasebi et al., Cell Stem Cell (2014) and added in Figure 4—figure supplement 1B-1C. We added the description for this result in line 492-497, page 26 in the Results.

Additionally, the individual curves of all 6 wildtype and 6 mutant mice, which are used for statistical analysis in Figure 4C-4D, were shown in Figure 4—figure supplement 1A.

2. Consider a more simple tethering assay allowing for simpler interpretation and a more convincing conclusion.

According to the editor’s and reviewer’s suggestion, we added a simpler strategy in HEK293 cells, in which we used the MS2-binding motif to recruit PTBP2 to exogenous FGF13 mRNA. As shown in Figure 6C, the MS2-binding motif was inserted into the 5’-UTR of FGF13 mRNA. This MS2-FGF13 mRNA exhibited an increased binding to MS2 coat protein (MCP)-fused PTBP2, which was revealed by the RIP experiment (see Figure 6D). Moreover, recruitment of PTBP2 to FGF13 mRNA could increase the protein level of FGF13 (see Figure 6E). Therefore, PTBP2 recruitment is sufficient to drive *FGF13* translation with this simpler experiment. We now added this result in Figure 6C-6E, and described it in line 669-677, page 35 in the Results, and line 863-864, page 45 in the Discussion.

3. Perform RIP type experiments in iPSCs to establish interaction between SNP and endogenous PTBP2.

According to the editor’s and reviewer’s suggestion, we used wildtype and mutant iPSCs, which were also used in Figure 1J-1K, to verify the altered endogenous interaction. As shown in Figure 5J-5L, the RIP experiment showed a reduced enrichment of mutant FGF13 mRNA compared with wildtype FGF13 mRNA through IP with PTBP2 antibody. This result further suggests that the 5'-UTR SNP of FGF13 largely decreases its interaction with endogenous PTBP2. We now added this result in Figure 5J-5L, and described it in line 604-609, page 31 in the Results.

4, Provide better graphic representation of the cellular phenotyping data (i.e. additional images, higher magnification, additional parameters measured)

According to the editor’s and reviewer’s suggestion, we now replaced and added more representative images of neurons in Figure 1—figure supplement 1 and in Figure 3C. Besides, we also added the parameter of total axon length in Figure 1L. We added the description for this result in line 240, page 12 in the Results.

Reviewer #1:In this work, Pan et al. seek to expand our understanding of protein expression and how this contributes to human disease. The authors focus on FGF13 because of recent advancements suggesting its involvement in neural development and identify a C≥G SNP in the 5'UTR of FGF13 which is found in 3 male ID patients and is therefore enriched in the ID patient population relative to the general population (~2% compared to ~0.1-0.3%). Based on whole-genome sequencing of the affected individuals, the authors posit that this SNP may be causal for ID. The rest of the work takes this SNP into cell culture and mouse models in order to understand how it contributes to the ID phenotype. The authors observe modest isoform-specific reductions in protein levels of FGF13 in SNP-harboring HEK293 cells, iPSCs from patients, iPSC-derived neurons, and in mouse neural tissue. Mice feature slight behavioral and memory impairments, and slight defects in neural migration, polarization, and branching. Polysome profiling and ribosome-IP experiments suggest that the SNP slightly decreases FGF13 mRNA association with translating ribosomes. Mass spectrometry identifies PTBP1 and PTBP2 as proteins which preferentially associate with the WT 5'UTR of FGF13 over the SNP-containing 5'UTR. The authors show that decreasing PTBP expression also decreases FGF13 expression, and a tethering assay is attempted to show that recruitment of PTBP2 to the FGF13 5'UTR drives expression. Together, this combination of data leads the authors to propose a model whereby the C≥G SNP disrupts PTBP2 binding to the FGF13 5'UTR, leading to decreased FGF13 protein expression, disrupted neural development, and ultimately the ID phenotype.While this work presents an intriguing model for regulation of FGF13, there are several key concerns that limit enthusiasm for publication at this stage.1) The effect sizes throughout the cell culture and mouse model experiments are small. Slight decreases are observed in FGF13 protein expression and in neuronal migration and branching, and the mouse behavioral and memory impairments appear to be small and unenlightening due to the random assortment of positive and negative behavioral assays. These modest effect sizes stand in stark contrast to the severe ID phenotypes observed in the affected patients (IQ < 40, significant deficiencies in adaptive behaviors). This is concerning and calls to question whether the C≥G SNP is actually causal for the ID phenotype, or whether the mouse is an appropriate model to study the FGF13 5'UTR despite demonstrated conservation at the SNP site. A discussion of ID penetrance for humans harboring this SNP is warranted since ~0.1-0.3% of the population appears to carry this SNP.

Nowadays, mouse models provide valuable clues for the function and mechanism research of human neural diseases. In the present study, we also used human cellular models directly from patients to test the function of the SNP, and the results were consistent with our animal studies (at molecular, cellular and behavior levels). We agree with the reviewer that the effect sizes in cellular model and mouse model were modest. The studies are still required to correlate the degree of changes at cell culture and mouse models with the patients suffering from human neural diseases. Although the gap exists between human and mouse in their development process, the mouse is still a useful model for human neural diseases. Actually, the development of human brain is more complicated, and some human neural diseases have more severe phenotypes than those in mice. One example is doublecortin (DCX), which is a classical microtubule-associated protein. Patients with mutant DCX have lissencephaly and double cortex syndrome (Gleeson et al., 1998), while the loss of DCX in rats or mice only results in increased axon branching and delayed neuronal migration (Bai et al., 2003; Kappeler et al., 2006). Now we added the discussion in line 895-899, page 47 in the Discussion.

According to the reviewer’s comment, we have modified the description “In three families of the present study, all male carriers exhibited ID phenotypes. However, whether the allele frequency in general population (~0.2%) comes from unaffected female carriers or from reduced penetrance of male carriers is still unknown. The true penetrance of the SNP in general population could only be estimated based on data from more families of affected individuals.” in line 796-801, page 42 in the Discussion.

2) The evidence for translational regulation is very modest – the small redistribution of FGF13 mRNA on the polysome gradient to a lighter fraction in Figure 4 seems unlikely to report on the ~25% decrease in protein expression reported in Figure 2D. This assay can be a useful one for looking at translational regulation but a greater shift would be more compelling (see Figure 2C in Tahmasebi et al., Cell Stem Cell (2014) for an example of a more compelling shift).

According to the reviewer’s suggestion, we performed recommended assay to further look at translational regulation. The semi-quantitative PCR following the polysome profiling from 3 wildtype and 3 mutant mice also detected that FGF13 mRNA showed less association with heavy polysomes in mutant mice. This result was shown in the format of that in Tahmasebi et al., Cell Stem Cell (2014) and added in Figure 4—figure supplement 1B-1C.

3) The authors model is predicated on a role of the PTBPs as enhancers of mRNA translation, whereas prior evidence suggests the major role of PTBPs to be in splicing. While this would be a novel and interesting role for PTBPs, the authors do not adequately discuss the known functions of PTBPs or canonical PTBP binding motifs, and do little experimentally to convince the reader of this novel function (e.g. subcellular localization). Finally, the experimental setup and interpretations of the tethering assay (Figure 6) are convoluted and provide no support for their model. Tethering is a valuable tool as it obviates the need for a well-defined binding site, but that is the way to utilize it – recruitment of PTBP2 to an orthogonal site such as an MS2 binding motif would be a cleaner method to show that PTBP recruitment is sufficient to drive translation of the FGF13 mRNA.

According to the reviewer’s suggestion, we added a simpler strategy in HEK293 cells, in which we used the MS2-binding motif to recruit PTBP2 to exogenous FGF13 mRNA. As shown in Figure 6C, the MS2-binding motif was inserted into the 5'-UTR of FGF13 mRNA. This MS2-FGF13 mRNA exhibited an increased binding to MS2 coat protein (MCP)-fused PTBP2, which was revealed by the RIP experiment (see Figure 6D). Moreover, recruitment of PTBP2 to FGF13 mRNA could increase the protein level of FGF13 (see Figure 6E). Therefore, PTBP2 recruitment is sufficient to drive *FGF13* translation with this simpler experiment. We now added this result in Figure 6C-6E, and described it in line 669-677, page 35 in the Results, and line 863-864, page 45 in the Discussion. The original PUF experiment is used to rescue the disrupted cellular phenotypes in mutant neurons and still kept in Results.

We further performed the biochemical assay of nucleus-cytoplasm isolation using cultured cortical neurons. Immunoblotting showed that PTBP2 was distributed both in the cytoplasm and nucleus, similar to the previous result in mouse testes [Figure 3C, (Xu and Hecht, 2007)]. We now added this result in Figure 6—figure supplement 1, and described it in line 663-664, page 35 in the Results.

In the revised manuscript, we have also modified the description “PTBP1/2 usually play major roles in regulating pre-mRNA splicing by binding to pyrimidine-rich sequences. They could also affect miRNA function and cytoplasmic translation (Sawicka et al., 2008; Kafasla et al., 2012). Our study provides further evidence for the cytoplasmic function of PTBPs in FGF13 translation.” in line 870-873, page 46 in the Discussion.

Reviewer #3:The authors demonstrate that a C to G variant in the 5'UTR of the FGF13 gene results in decreased binding of an RNA binding protein PTBP2 which appears to affect translation was reduced. An increase of the binding between PTBP2 and mutant FGF13 5'UTR with a mutant PUF-PTBP2 rescue translation. The study also highlights a potentially interesting role for PTBP2 to enhance translation by 5'UTR binding, albeit it is not clear how that happens. The study has many positives, particularly the use of orthogonal systems (IPSC lines/neurons and mouse models) to evaluate the molecular, cellular and also behavioral impact of the 5'UTR mutation. The methods used are standard for the field, but nevertheless serve as a logical framework for tackling such a question. The prevalence of the mutation is low and while the mechanism described/revealed in this paper is interesting and highlights the relevance of non-coding mutations on translation control, particularly FGF13, in impaired learning and memory, it's difficult to conclude that that's the same as intellectual disability.The writing in the paper is poor and there are many spelling errors and grammatical mistakes through the figures, figure legends and text. It really distracts from the science, which in contrast is actually expansion and interesting, using a range of orthogonal in vitro and in vivo approaches to identify the mechanism and consequences underlying a point mutation found in the 5'UTR of FGF13, suggested to impact protein translation of the gene.1. Figure 1L. % of polarized neurons do not look convincing from the IHC images.

The ratio of polarized neurons was determined by analyzing a group of neurons. Now, we labeled unpolarized neurons with arrow in Figure 1L, and also added more representative images of polarized neurons derived from WT and Mut iPSCs in Figure 1—figure supplement 1.

2. Figure 4 is not convincing (4a, 4b) as well.

According to the editor’s and reviewer’s comments, we performed further experiment to verify this result of Figure 4B. The semi-quantitative PCR following the polysome profiling from 3 wildtype and 3 mutant mice also detected that FGF13 mRNA showed less association with heavy polysomes in mutant mice. This result was shown as suggested in the format of that in Tahmasebi et al., Cell Stem Cell (2014) and added in Figure 4—figure supplement 1B-1C. We added the description for this result in line 492-497, page 26 in the Results.

Additionally, the individual curves of all 6 wildtype and 6 mutant mice, which are used for statistical analysis in Figure 4C-4D, were shown in Figure 4—figure supplement 1A.

3. PTBP2 knockout mice are available and phenotypes have been previously scored – can the authors discuss/make a comparison – is FGF13 isoform 4 levels lower in those mice?

*Ptbp2* null mice (*EIIa*-Cre knockout) and *Ptbp2 Nestin*-Cre knockout mice (*Ptbp2*-*Nes*KO) all died at birth due to apparent respiratory failure (Lakso et al., 1996; Tronche et al., 1999). They exhibited normal morphology for major brain structures at late embryonic stage E18.5, but the axonal tracts in white matters were absent (Li et al., 2014). *Ptbp2Emx1*-Cre knockout mice (*Ptbp2*-*Emx*KO) were variable and similar to control pups at P0, but displayed slower growth with smaller size and lower body weight during the first three postnatal weeks, and died around P21. The cortex of *Ptbp2*-*Emx*KO displayed widespread neuronal death and degeneration, and was substantially thinner than control cortex (Li et al., 2014). In contrast, *Fgf13Emx1*-Cre knockout mice (*Fgf13*-*Emx*KO) exhibited normal size, could survive to adult stage, and had similar brain structures compared to controls (Wu et al., 2012). The overall phenotypes of *Fgf13* knockout mice are milder than *Ptbp2* knockout mice, because PTBP2 is known to regulate many target transcripts and influence extensive biological processes, while FGF13 stabilizes microtubules and regulates neuronal polarity, axon branching and neuronal migration (Wu et al., 2012).

Additionally, previous studies using *Ptbp2* knockout mice exclusively focus on its role of alternative splicing and the targeted transcripts. We did not find the data that *Fgf13* was directly targeted by PTBP2 or splicing changes of *Fgf13* was detected from the CLIP-seq data and RNA-seq data of *Ptbp2* knockout mice (Licatalosi et al., 2012; Li et al., 2014).

4. The authors should have in the main figures EMSAs to show binding to the RNA and the mutant by PTBP2/PTBP1.

The biotin-labeled RNA pull-down assay was used in our study to detect the binding alteration caused by mutation. EMSAs could also be used to study the interaction between RNAs and RNA-binding proteins. We are truly sorry that we are not able to finish this experiment, because we could not get the isotopes due to the influence of COVID-19.

5. Can the authors show by CLIP/UV-RIP the binding of PTBP1/2 to the endogenous wildtype and mutant RNA?

According to the reviewer’s suggestion, we used wildtype and mutant iPSCs, which were also used in Figure 1J-1K, to verify the altered endogenous interaction. As shown in Figure 5J-5L, the RIP experiment showed a reduced enrichment of mutant FGF13 mRNA compared with wildtype FGF13 mRNA through IP with PTBP2 antibody. This result further suggests that the 5'-UTR SNP of FGF13 largely decreases its interaction with endogenous PTBP2. We now added this result in Figure 5J-5L, and described it in line 604-609, page 31 in the Results.

References:

Bai J, Ramos RL, Ackman JB, Thomas AM, Lee RV, LoTurco JJ. 2003. RNAi reveals doublecortin is required for radial migration in rat neocortex. Nat Neurosci 6: 1277-1283. doi: https://doi.org/10.1038/nn1153, PMID: 14625554

Gleeson JG, Allen KM, Fox JW, Lamperti ED, Berkovic S, Scheffer I, Cooper EC, Dobyns WB, Minnerath SR, Ross ME, Walsh CA. 1998. Doublecortin, a brain-specific gene mutated in human X-linked lissencephaly and double cortex syndrome, encodes a putative signaling protein. Cell 92: 63-72. doi: https://doi.org/10.1016/s0092-8674(00)80899-5, PMID: 9489700

Kafasla P, Mickleburgh I, Llorian M, Coelho M, Gooding C, Cherny D, Joshi A, Kotik-Kogan O, Curry S, Eperon IC, Jackson RJ, Smith CW. 2012. Defining the roles and interactions of PTB. Biochem Soc Trans 40: 815-820. doi: https://doi.org/10.1042/bst20120044, PMID: 22817740

Kappeler C, Saillour Y, Baudoin JP, Tuy FP, Alvarez C, Houbron C, Gaspar P, Hamard G, Chelly J, Metin C, Francis F. 2006. Branching and nucleokinesis defects in migrating interneurons derived from doublecortin knockout mice. Hum Mol Genet 15: 1387-1400. doi: https://doi.org/10.1093/hmg/ddl062, PMID: 16571605

Lakso M, Pichel JG, Gorman JR, Sauer B, Okamoto Y, Lee E, Alt FW, Westphal H. 1996. Efficient in vivo manipulation of mouse genomic sequences at the zygote stage. Proc Natl Acad Sci U S A 93: 5860-5865. doi: https://doi.org/10.1073/pnas.93.12.5860, PMID: 8650183

Li Q, Zheng S, Han A, Lin CH, Stoilov P, Fu XD, Black DL. 2014. The splicing regulator PTBP2 controls a program of embryonic splicing required for neuronal maturation. *eLife* 3: e01201. doi: https://doi.org/10.7554/*eLife*.01201, PMID: 24448406

Licatalosi DD, Yano M, Fak JJ, Mele A, Grabinski SE, Zhang C, Darnell RB. 2012. Ptbp2 represses adult-specific splicing to regulate the generation of neuronal precursors in the embryonic brain. Genes Dev 26: 1626-1642. doi: https://doi.org/10.1101/gad.191338.112, PMID: 22802532

Sawicka K, Bushell M, Spriggs KA, Willis AE. 2008. Polypyrimidine-tract-binding protein: a multifunctional RNA-binding protein. Biochem Soc Trans 36: 641-647. doi: https://doi.org/10.1042/BST0360641, PMID: 18631133

Tronche F, Kellendonk C, Kretz O, Gass P, Anlag K, Orban PC, Bock R, Klein R, Schütz G. 1999. Disruption of the glucocorticoid receptor gene in the nervous system results in reduced anxiety. Nat Genet 23: 99-103. doi: https://doi.org/10.1038/12703, PMID: 10471508

Wu QF, Yang L, Li S, Wang Q, Yuan XB, Gao X, Bao L, Zhang X. 2012. Fibroblast growth factor 13 is a microtubule-stabilizing protein regulating neuronal polarization and migration. Cell 149: 1549-1564. doi: https://doi.org/10.1016/j.cell.2012.04.046, PMID: 22726441

Xu M, Hecht NB. 2007. Polypyrimidine tract binding protein 2 stabilizes phosphoglycerate kinase 2 mRNA in murine male germ cells by binding to its 3'UTR. Biol Reprod 76: 1025-1033. doi: https://doi.org/10.1095/biolreprod.107.060079, PMID: 17329592